# HOMOTOPY RELAXATION TRAINING ALGORITHMS FOR INFINITE-WIDTH TWO-LAYER ReLU NEURAL NETWORKS

## ABSTRACT

In this paper, we present a novel training approach called the Homotopy Relaxation Training Algorithm (HRTA), aimed at accelerating the training process in contrast to traditional methods. Our algorithm incorporates two key mechanisms: one involves building a homotopy activation function that seamlessly connects the linear activation function with the ReLU activation function; the other technique entails relaxing the homotopy parameter to enhance the training refinement process. We have conducted an in-depth analysis of this novel method within the context of the neural tangent kernel (NTK), revealing significantly improved convergence rates. Our experimental results, especially when considering networks with larger widths, validate the theoretical conclusions. This proposed HRTA exhibits the potential for other activation functions and deep neural networks.

## 1 INTRODUCTION

Neural networks (NNs) have become increasingly popular and widely used in scientific and engineering applications, such as image classification Krizhevsky et al. (2017); He et al. (2015), regularization He et al. (2016); Wu et al. (2018). Finding an efficient way to train and obtain the parameters in NNs is an important task, enabling the application of NNs in various domains.

Numerous studies have delved into training methods for NNs, as evidenced by the works of Erhan et al. (2010), Keskar et al. (2016), and You et al. (2019) Erhan et al. (2010); Keskar et al. (2016); You et al. (2019). However, the optimization of loss functions can become increasingly challenging over time, primarily due to the nonconvex energy landscape. Traditional algorithms such as the gradient descent method and the Adam method often lead to parameter entrapment in local minima or saddle points for prolonged periods. The homotopy training algorithm (HTA) was introduced as a remedy by making slight modifications to the NN structure. HTA draws its roots from the concept of homotopy continuation methods Morgan & Sommese (1987); Sommese & Wampler (2005); Hao (2018, 2022), with its initial introduction found in Chen & Hao (2019). However, constructing a homotopy function requires it to be aligned with the structure of neural networks and entails time-consuming training.

In this paper, we introduce an innovative training approach called the homotopy relaxation training algorithm (HRTA). This approach leverages the homotopy concept, specifically focusing on the activation function, to address the challenges posed by the HRTA. We develop a homotopy activation function that establishes a connection between the linear activation function and the target activation function. By gradually adjusting the homotopy parameter, we enable a seamless transition toward the target activation function. Mathematically, the homotopy activation function is defined as $\sigma_s$, where $s$ is the homotopy parameter, and it takes the form $\sigma_s(x) = (1 - s)\text{Id}(x) + s\sigma(x)$. Here, $\text{Id}(x)$ represents the identity function (i.e., the linear activation function), and $\sigma(x)$ is the target activation function. The term "homotopy" in the algorithm signifies its evolution from an entirely linear neural network, where the initial activation function is the identity function ($s = 0$). The homotopy activation function undergoes gradual adjustments until it transforms into a target neural network ($s = 1$). This transition, from the identity function to the target function, mirrors the principles of homotopy methods. Furthermore, our analysis reveals that by extrapolating (or over-relaxing) the homotopy parameter ($1 < s < 2$), training performance can be further enhanced. In

this context, we extend the concept of homotopy training, introducing what we refer to as "homotopy relaxation training."

In this paper, we relax the homotopy parameter and allow $s$ to take on any positive value in $[0, 2]$, rather than being restricted to values in $[0, 1]$. Moreover, we provide theoretical support for this algorithm, particularly in hyperparameter scenarios. Our analysis is closely related to neural tangent kernel methods Jacot et al. (2018); Arora et al. (2019); Zhang et al. (2020); Cao & Gu (2020); Du et al. (2018); Huang et al. (2023). We establish that modifying the homotopy parameter at each step increases the smallest eigenvalue of the gradient descent kernel for infinite-width neural networks (see Theorem 1). Consequently, we present Theorem 2 to demonstrate the capacity of HRTA to enhance training speed.

The paper is organized as follows: We introduce the HRTA in Section 2. Next, in Section 4, we conduct experiments, including supervised learning and solving partial differential equations, based on our algorithm. Finally, in Section 3, we provide theoretical support for our theory.

## 2 HOMOTOPY RELAXATION TRAINING ALGORITHM

In this paper, we consider supervised learning for NNs. Within a conventional supervised learning framework, the primary objective revolves around acquiring an understanding of a high-dimensional target function denoted as $f(\boldsymbol{x})$, which finds its domain in $(0, 1)^d$, with $\|f\|_{L^\infty((0,1)^d)} \leq 1$, through a finite collection of data samples $\{(\boldsymbol{x}_i, f(\boldsymbol{x}_i))\}_{i=1}^n$. When embarking on the training of a NN, our aim rests upon the discovery of a NN representation denoted as $\phi(\boldsymbol{x}; \boldsymbol{\theta})$ that serves as an approximation to $f(\boldsymbol{x})$, a feat achieved through the utilization of randomly gathered data samples $\{(\boldsymbol{x}_i, f(\boldsymbol{x}_i))\}_{i=1}^n$, with $\boldsymbol{\theta}$ representing the parameters within the NN architecture. It is assumed, in this paper, that the sequence $\{\boldsymbol{x}_i\}_{i=1}^n$ constitutes an independent and identically distributed (i.i.d.) sequence of random variables, uniformly distributed across $(0, 1)^d$. Denote

$$\boldsymbol{\theta}_S := \arg\min_{\boldsymbol{\theta}} \mathcal{R}_S(\boldsymbol{\theta}) := \arg\min_{\boldsymbol{\theta}} \frac{1}{2n} \sum_{i=1}^n |f(\boldsymbol{x}_i) - \phi(\boldsymbol{x}_i; \boldsymbol{\theta})|^2. \tag{1}$$

Next, we introduce the HRTA by defining $\sigma_{s_p}(x) = (1 - s_p)\mathrm{Id}(x) + s_p\sigma(x)$ with a discretized set points of the homotopy parameter $\{s_p\}_{p=1}^M \in (0, 2)$. We then proceed to obtain:

$$\boldsymbol{\theta}_S^{s_p} := \arg\min_{\boldsymbol{\theta}} \mathcal{R}_{S,s_p}(\boldsymbol{\theta}), \text{ with an initial guess } \boldsymbol{\theta}_S^{s_{p-1}}, p = 1, \cdots, M, \tag{2}$$

where we initialize $\boldsymbol{\theta}_S^{s_0}$ randomly, and $\boldsymbol{\theta}_S^{s_M}$ represents the optimal parameter value that we ultimately achieve.

In this paper, we consider a two-layer NN defined as follows

$$\phi(\boldsymbol{x}; \boldsymbol{\theta}) := \frac{1}{\sqrt{m}} \sum_{k=1}^m a_k \sigma(\boldsymbol{\omega}_k^\mathsf{T} \boldsymbol{x}), \tag{3}$$

with the activation function $\sigma(z) = \mathrm{ReLU}(z) = \max\{z, 0\}$. The evolution of the traditional training can be written as the following differential equation:

$$\frac{\mathrm{d}\boldsymbol{\theta}(t)}{\mathrm{d}t} = -\nabla_{\boldsymbol{\theta}} \mathcal{R}_S(\boldsymbol{\theta}(t)). \tag{4}$$

In the HRTA setup, we train a sequences of leaky ReLU activate functions Xu et al. (2020); Mastro-michalakis (2020). Subsequently, for each of these Leaky ReLUs with given $s_p$, we train the neural network on a time interval of $[t_{p-1}, t_p]$:

$$\frac{\mathrm{d}\boldsymbol{\theta}(t)}{\mathrm{d}t} = -\nabla_{\boldsymbol{\theta}} \mathcal{R}_{S,s_p}(\boldsymbol{\theta}(t)). \tag{5}$$

Moreover, we have $t_0 = 0$ and initialize the parameter vector $\boldsymbol{\theta}(0)$, drawn from a normal distribution $\mathcal{N}(\mathbf{0}, \boldsymbol{I})$. Therefore the HRTA algorithm's progression is outlined in **Algorithm 1**.

**Remark 1.** *If $s_M = 1$, then upon completion, we will have obtained $\boldsymbol{\theta}(t_M)$ and the NN $\phi_{s_M}(\boldsymbol{x}; \boldsymbol{\theta}(t_M))$, characterized by pure ReLU activations. The crux of this algorithm is its ability to transition from Leaky ReLUs to a final configuration of a NN with pure ReLU activations. This transformation is orchestrated via a series of training iterations utilizing the homotopy approach.*

*However, our paper demonstrates that there is no strict necessity to achieve $s_M = 1$. What we aim for is to obtain a NN with parameters $\boldsymbol{\theta}$ that minimizes $\mathcal{R}_{S,s_M}(\boldsymbol{\theta})$. This is because, for any value of $s$, we can readily represent $\phi_s(\boldsymbol{x}; \boldsymbol{\theta})$ as a pure ReLU NN, as shown in the following equation:*

$$\sigma_s(x) = (1-s)Id(x) + s\sigma(x) = (1-s)\sigma(x) - (1-s)\sigma(-x) + s\sigma(x) = \sigma(x) - (1-s)\sigma(-x).$$

*To put it simply, if we can effectively train a NN with Leaky-ReLU to learn the target functions, it implies that we can achieve the same level of performance with a NN using standard ReLU activation. Consequently, the theoretical analysis in the paper does not require that the final value of $s_M$ must be set to 1. Moreover, our method is applicable even when $s_M > 1$, which we refer to as the relaxation part of HRTA. It's important to highlight that for $s > 1$ the decay speed may surpass that of a pure ReLU neural network. This is consistent with the training using the hat activation function (specifically, when $s = 2$ in the homotopy activation function) discussed in Hong et al. (2022). , although it's worth noting that their work primarily focuses on the linear case (involving only the constant factor change), whereas our work extends this consideration to neural networks.*

---

**Algorithm 1:** The Homotopy Relaxation Training Algorithm for Two Layer Neural Netwroks

**input** : Sample points of function $\{(\boldsymbol{x}_i, f(\boldsymbol{x}_i))\}_{i=1}^{n}$; Initialized homotopy parameter $s_1 > 0$;
Number of the iteration times $M$; $\zeta > 0$; Training time of each iteration $T$; $t_p = pT$;
learning rate $\tau$; $\boldsymbol{\theta}_0 \sim \mathcal{N}(0, \boldsymbol{I})$.

1 **for** $p = 1, 2, \ldots, M$ **do**
2     **for** $t \in [t_{p-1}, t_p]$ **do**
3        $\boldsymbol{\theta}_{t+1} = \boldsymbol{\theta}_t - \tau\nabla_{\boldsymbol{\theta}}(\mathcal{R}_{s_p}(\boldsymbol{\theta}_t))$;
4     **end**
5     $s_{p+1} := s_p + \zeta$;
6 **end**

**output:** $\phi_{s_M}(\boldsymbol{x}, \boldsymbol{\theta}_{T_M})$ as the two layer NN approximation to approximate $f(\boldsymbol{x})$.

---

## 3 CONVERGENCE ANALYSIS

In this section, we will delve into the convergence analysis of the HRTA based on the neural target kernel methods Jacot et al. (2018); Arora et al. (2019); Zhang et al. (2020); Cao & Gu (2020); Du et al. (2018); Huang et al. (2023). For simplicity, we will initially focus on the case where $M = 2$. It is important to note that for cases with $M > 2$, all the analyses presented here can be readily extended. To start, we set the initial value of $s_1 > 0$. The structure of the proof of Theorem 2 is shown in Figure 1.

### 3.1 GRADIENT DESCENT KERNEL

The kernels characterizing the training dynamics for the $p$-th iteration take the following form:

$$k_p^{[a]}(\boldsymbol{x}, \boldsymbol{x}') := \mathbf{E}_{\boldsymbol{\omega}}\sigma_{s_p}(\boldsymbol{\omega}^{\mathsf{T}}\boldsymbol{x})\sigma_{s_p}(\boldsymbol{\omega}^{\mathsf{T}}\boldsymbol{x}'), \quad k_p^{[\boldsymbol{\omega}]}(\boldsymbol{x}, \boldsymbol{x}') := \mathbf{E}_{(a,\boldsymbol{\omega})}a^2\sigma'_{s_p}(\boldsymbol{\omega}^{\mathsf{T}}\boldsymbol{x})\sigma'_{s_p}(\boldsymbol{\omega}^{\mathsf{T}}\boldsymbol{x}')\boldsymbol{x}\cdot\boldsymbol{x}'. \quad (6)$$

The Gram matrices, denoted as $\boldsymbol{K}_p^{[a]}$ and $\boldsymbol{K}_p^{[\boldsymbol{\omega}]}$, corresponding to an infinite-width two-layer network with the activation function $\sigma_{s_p}$, can be expressed as follows:

$$K_{ij,p}^{[a]} = k_p^{[a]}(\boldsymbol{x}_i, \boldsymbol{x}_j), \ \boldsymbol{K}_p^{[a]} = (K_{ij,p}^{[a]})_{n\times n}, \ K_{ij,p}^{[\boldsymbol{\omega}]} = k_p^{[\boldsymbol{\omega}]}(\boldsymbol{x}_i, \boldsymbol{x}_j), \ \boldsymbol{K}_p^{[\boldsymbol{\omega}]} = (K_{ij,p}^{[\boldsymbol{\omega}]})_{n\times n} \quad (7)$$

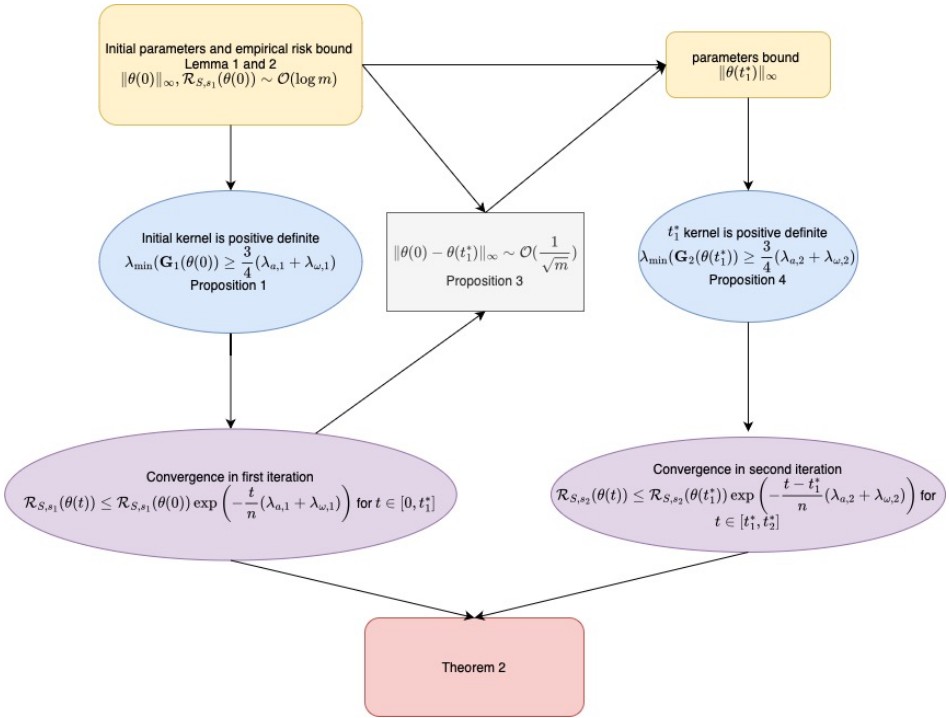

Figure 1: Structure of proof of Theorem 2

Moreover, the Gram matrices, referred to as $\boldsymbol{G}_p^{[a]}$ and $\boldsymbol{G}_p^{[\boldsymbol{\omega}]}$, corresponding to a finite-width two-layer network with the activation function $\sigma_{s_p}$, can be defined as

$$G_{ij,p}^{[a]} = \frac{1}{m} \sum_{k=1}^{m} \sigma_{s_p}(\boldsymbol{\omega}_k^\mathsf{T} \boldsymbol{x}) \sigma_{s_p}(\boldsymbol{\omega}_k^\mathsf{T} \boldsymbol{x}'),\ \boldsymbol{G}_p^{[a]} = (G_{ij,p}^{[a]})_{n \times n},$$

$$G_{ij,p}^{[\boldsymbol{\omega}]} = \frac{1}{m} \sum_{k=1}^{m} a^2 \sigma'_{s_p}(\boldsymbol{\omega}_k^\mathsf{T} \boldsymbol{x}) \sigma'_{s_p}(\boldsymbol{\omega}_k^\mathsf{T} \boldsymbol{x}') \boldsymbol{x} \cdot \boldsymbol{x}',\ \boldsymbol{K}_p^{[\boldsymbol{\omega}]} = (G_{ij,p}^{[\boldsymbol{\omega}]})_{n \times n}. \tag{8}$$

**Assumption 1.** *Denote $\boldsymbol{K}^{[a]}$ and $\boldsymbol{K}^{[\boldsymbol{\omega}]}$ are the Gram matrices for ReLU neural network and*

$$H_{ij}^{[a]} = k^{[a]}(-\boldsymbol{x}_i, \boldsymbol{x}_j),\ \boldsymbol{H}^{[a]} = (H_{ij}^{[a]})_{n \times n},\ H_{ij}^{[\boldsymbol{\omega}]} = k^{[\boldsymbol{\omega}]}(-\boldsymbol{x}_i, \boldsymbol{x}_j),\ \boldsymbol{H}^{[\boldsymbol{\omega}]} = (H_{ij}^{[\boldsymbol{\omega}]})_{n \times n}$$

$$M_{ij}^{[a]} = k^{[a]}(\boldsymbol{x}_i, -\boldsymbol{x}_j),\ \boldsymbol{M}^{[a]} = (M_{ij}^{[a]})_{n \times n},\ M_{ij}^{[\boldsymbol{\omega}]} = k^{[\boldsymbol{\omega}]}(\boldsymbol{x}_i, -\boldsymbol{x}_j),\ \boldsymbol{M}^{[\boldsymbol{\omega}]} = (M_{ij}^{[\boldsymbol{\omega}]})_{n \times n}$$

$$T_{ij}^{[a]} = k^{[a]}(-\boldsymbol{x}_i, -\boldsymbol{x}_j),\ \boldsymbol{T}^{[a]} = (T_{ij}^{[a]})_{n \times n},\ T_{ij}^{[\boldsymbol{\omega}]} = k^{[\boldsymbol{\omega}]}(-\boldsymbol{x}_i, -\boldsymbol{x}_j),\ \boldsymbol{T}^{[\boldsymbol{\omega}]} = (T_{ij}^{[\boldsymbol{\omega}]})_{n \times n}. \tag{9}$$

*Suppose that all matrices defined above are strictly positive definite.*

**Remark 2.** *We would like to point out that if, for all $i$ and $j$ satisfying $i \neq j$, we have $\pm\boldsymbol{x}_i$ not parallel to $\pm\boldsymbol{x}_j$, then Assumption 1 is satisfied. The validity of this assertion can be established by referring to (Du et al., 2018, Theorem 3.1) for the proof.*

**Theorem 1.** *Suppose Assumption 1 holds, denote*

$$\lambda_{a,p} := \lambda_{min}\left(\boldsymbol{K}_p^{[a]}\right),\ \lambda_{\boldsymbol{\omega},p} := \lambda_{min}\left(\boldsymbol{K}_p^{[\boldsymbol{\omega}]}\right).$$

*Then we have $\lambda_{\boldsymbol{\omega},p+1} \geq \lambda_{\boldsymbol{\omega},p} > 0$, $\lambda_{a,p+1} \geq \lambda_{a,p} > 0$ for all $0 \leq s_p \leq s_{p+1}$.*

## 3.2 CONVERGENCE OF $t_1$ ITERATION

All the proofs for this subsection can be found in Appendix A.3.

**Lemma 1** (bounds of initial parameters). *Given $\delta \in (0,1)$, we have with probability at least $1-\delta$ over the choice of $\boldsymbol{\theta}(0)$ such that*

$$\max_{k\in[m]}\{|a_k(0)|, \|\boldsymbol{\omega}_k(0)\|_\infty\} \leq \sqrt{2\log\frac{2m(d+1)}{\delta}}. \tag{10}$$

Next we are plan to bound $\mathcal{R}_{S,s_1}(\boldsymbol{\theta}(0))$ based on lemmas of Rademacher complexity.

**Lemma 2** (bound of initial empirical risk). *Given $\delta \in (0,1)$ and the sample set $S = \{(\boldsymbol{x}_i, y_i)\}_{i=1}^n \subset \Omega$ with $\boldsymbol{x}_i$ 's drawn i.i.d. from uniform distribution. Suppose that Assumption 1 holds. We have with probability at least $1-\delta$ over the choice of $\boldsymbol{\theta}(0)$*

$$\mathcal{R}_{S,s_1}(\boldsymbol{\theta}(0)) \leq \frac{1}{2}\left[1 + 2d\log\frac{4m(d+1)}{\delta}\left(2 + 6\sqrt{2\log(8/\delta)}\right)\right]^2. \tag{11}$$

Moving forward, we will now delineate the core of the homotopy relaxation training process within each iteration. Due to the distinct nature of $\boldsymbol{\omega}_k$ and $a_k$, the training dynamics for the $p$-th iteration can be expressed as follows:

$$\begin{cases} \frac{\mathrm{d}a_k(t)}{\mathrm{d}t} = -\nabla_{a_k}\mathcal{R}_{S,s_1}(\boldsymbol{\theta}) = -\frac{1}{n\sqrt{m}}\sum_{i=1}^n e_{i,1}\sigma_{s_p}\left(\boldsymbol{w}_k^\top \boldsymbol{x}_i\right) \\ \frac{\mathrm{d}\boldsymbol{\omega}_k(t)}{\mathrm{d}t} = -\nabla_{\boldsymbol{w}_k}\mathcal{R}_{S,s_1}(\boldsymbol{\theta}) = -\frac{1}{n\sqrt{m}}\sum_{i=1}^n e_{i,1}a_k\sigma'_{s_p}\left(\boldsymbol{w}_k^\top \boldsymbol{x}_i\right)\boldsymbol{x}_i \end{cases}$$

where $e_{i,1} = |f(\boldsymbol{x}_i) - \phi_{s_1}(\boldsymbol{x}_i; \boldsymbol{\theta})|$. Now, we are going to perform the convergence analysis. Before that, we need Proposition 1 to demonstrate that the kernel in the gradient descent dynamics during the initial phase is positive for finite-width NNs when $m$ is large.

**Proposition 1.** *Given $\delta \in (0,1)$ and the sample set $S = \{(\boldsymbol{x}_i, y_i)\}_{i=1}^n \subset \Omega$ with $\boldsymbol{x}_i$ 's drawn i.i.d. with uniformly distributed. Suppose that Assumption 1 holds. If $m \geq \frac{16n^2d^2C_{\psi,d}}{C_0\lambda^2}\log\frac{4n^2}{\delta}$ then with probability at least $1-\delta$ over the choice of $\boldsymbol{\theta}(0)$, we have*

$$\lambda_{\min}\left(\boldsymbol{G}_1\left(\boldsymbol{\theta}(0)\right)\right) \geq \frac{3}{4}(\lambda_{a,1} + \lambda_{\boldsymbol{\omega},1}).$$

Set

$$t_1^* = \inf\{t \mid \boldsymbol{\theta}(t) \notin \mathcal{N}_1(\boldsymbol{\theta}(0))\} \tag{12}$$

where

$$\mathcal{N}_1(\boldsymbol{\theta}(0)) := \left\{\boldsymbol{\theta} \mid \|\boldsymbol{G}_2(\boldsymbol{\theta}) - \boldsymbol{G}_1(\boldsymbol{\theta}(0))\|_F \leq \frac{1}{4}(\lambda_{a,1} + \lambda_{\boldsymbol{\omega},1})\right\}.$$

**Proposition 2.** *Given $\delta \in (0,1)$ and the sample set $S = \{(\boldsymbol{x}_i, y_i)\}_{i=1}^n \subset \Omega$ with $\boldsymbol{x}_i$ 's drawn i.i.d. with uniformly distributed. Suppose that Assumption 1 holds. If $m \geq \frac{16n^2d^2C_{\psi,d}}{C_0\lambda^2}\log\frac{4n^2}{\delta}$ then with probability at least $1-\delta$ over the choice of $\boldsymbol{\theta}(0)$, we have for any $t \in [0, t_1^*]$*

$$\mathcal{R}_{S,s_1}(\boldsymbol{\theta}(t)) \leq \mathcal{R}_{S,s_1}(\boldsymbol{\theta}(0))\exp\left(-\frac{t}{n}(\lambda_{a,1} + \lambda_{\boldsymbol{\omega},1})\right). \tag{13}$$

### 3.3 CONVERGENCE OF $t_2$ ITERATION

In this paper, without sacrificing generality, we focus our attention on the case where $M = 2$. However, it's important to note that our analysis and methodology can readily be extended to the broader scenario of $M \geq 2$. All the proof in this paper can be found in Appendix A.4

**Proposition 3.** *Given $\delta \in (0,1)$ and the sample set $S = \{(\boldsymbol{x}_i, y_i)\}_{i=1}^n \subset \Omega$ with $\boldsymbol{x}_i$ 's drawn i.i.d. with uniformly distributed. Suppose that Assumption 1 holds. If*

$$m \geq \max\left\{\frac{16n^2d^2C_{\psi,d}}{C_0\lambda^2}\log\frac{4n^2}{\delta}, \frac{8n^2d^2\mathcal{R}_{S,s_1}(\boldsymbol{\theta}(0))}{(\lambda_{a,1} + \lambda_{\boldsymbol{\omega},1})^2}\right\}$$

*then with probability at least $1-\delta$ over the choice of $\boldsymbol{\theta}(0)$, we have for any $t \in [0, t_1^*]$*

$$\max_{k\in[m]}\{|a_k(t) - a_k(0)|, \|\boldsymbol{\omega}_k(t) - \boldsymbol{\omega}_k(0)\|_\infty\} \leq \frac{8\sqrt{2}nd\sqrt{\mathcal{R}_{S,s_1}(\boldsymbol{\theta}(0))}}{\sqrt{m}(\lambda_{a,1} + \lambda_{\boldsymbol{\omega},1})}\sqrt{2\log\frac{4m(d+1)}{\delta}} =: \psi(m). \tag{14}$$

For simplicity, we define a $\mathcal{O}\left(\frac{\log m}{\sqrt{m}}\right)$ term $\psi$, which is

$$\psi(m) := \frac{8\sqrt{2}nd\sqrt{\mathcal{R}_{S,s_1}(\boldsymbol{\theta}(0))}}{(\lambda_{a,1} + \lambda_{\boldsymbol{\omega},1})}\sqrt{2\log\frac{4m(d+1)}{\delta}}.$$

Moving forward, we will employ $\boldsymbol{\theta}(t_1^*)$ as the initial value for training over $t_2$ iterations. However, before we proceed, it is crucial to carefully select the value of $s_2$. This choice of $s_2$ depends on both $\mathcal{R}_{S,s_1}(\boldsymbol{\theta}(t_1^*))$ and a constant $\zeta$, with the condition that $\zeta$ is a positive constant, ensuring that $0 < \zeta$. Therefore, we define $s_2$ as: $s_2 = s_1 + \zeta$ where $\zeta > 0$ is a constant.

It's important to emphasize that for each $\boldsymbol{\theta}(t_1^*)$, given that the training dynamics system operates without any random elements, we can determine it once we know $\boldsymbol{\theta}(0)$. In other words, we can consider $\boldsymbol{\theta}(t_1^*)$ as two distinct functions, $\bar{\boldsymbol{\theta}} = (\bar{a}, \boldsymbol{\omega})$, with $\boldsymbol{\theta}(0)$ as their input. This implies that $\bar{a}(\boldsymbol{\theta}(t_0)) = a(t_1^*)$ and $\bar{\boldsymbol{\omega}}(\boldsymbol{\theta}(0)) = \boldsymbol{\omega}(t_1^*)$.

**Proposition 4.** *Given $\delta \in (0, 1)$ and the sample set $S = \{(\boldsymbol{x}_i, y_i)\}_{i=1}^n \subset \Omega$ with $\boldsymbol{x}_i$ 's drawn i.i.d. with uniformly distributed. Suppose that Assumption 1 holds. If*

$$m \geq \max\left\{\frac{16n^2d^2C_{\psi,d}}{C_0\lambda^2}\log\frac{4n^2}{\delta}, n^4\left(\frac{128\sqrt{2}d\sqrt{\mathcal{R}_{S,s_1}(\boldsymbol{\theta}(0))}}{(\lambda_{a,1} + \lambda_{\boldsymbol{\omega},1})\min\{\lambda_{a,2}, \lambda_{\boldsymbol{\omega},2}\}}2\log\frac{4m(d+1)}{\delta}\right)\right\}$$

*then with probability at least $1 - \delta$ over the choice of $\boldsymbol{\theta}(0)$, we have*

$$\lambda_{\min}\left(\boldsymbol{G}_2(\boldsymbol{\theta}(t_1^*))\right) \geq \frac{3}{4}(\lambda_{a,2} + \lambda_{2,\boldsymbol{\omega}}).$$

**Remark 3.** *In accordance with Proposition 4, we can establish that $m$ follows a trend of $\mathcal{O}\left(\frac{\log(1/\delta)}{\min\{\lambda_{a,2}, \lambda_{\boldsymbol{\omega},2}\}}\right)$. This observation sheds light on our strategy of increasing the parameter $s$ with each iteration. As we have proven in Theorem 1, the smallest eigenvalues of Gram matrices tend to increase as $s$ increases.*

*Now, consider the second iteration. For a fixed value of $m$ that we have at this stage, a larger smallest eigenvalue implies that we can select a smaller value for $\delta$. Consequently, this leads to a higher probability of $\lambda_{\min}\left(\boldsymbol{G}_2(\boldsymbol{\theta}(t_1^*))\right)$ being positive.*

Set

$$t_2^* = \inf\{t \mid \boldsymbol{\theta}(t) \notin \mathcal{N}_2(\boldsymbol{\theta}(t_1^*))\} \tag{15}$$

where

$$\mathcal{N}_2(\boldsymbol{\theta}(t_1^*)) := \left\{\boldsymbol{\theta} \mid \|\boldsymbol{G}_2(\boldsymbol{\theta}) - \boldsymbol{G}_2(\boldsymbol{\theta}(t_1^*))\|_F \leq \frac{1}{4}(\lambda_{a,2} + \lambda_{\boldsymbol{\omega},2})\right\}.$$

**Proposition 5.** *Given $\delta \in (0, 1)$ and the sample set $S = \{(\boldsymbol{x}_i, y_i)\}_{i=1}^n \subset \Omega$ with $\boldsymbol{x}_i$ 's drawn i.i.d. with uniformly distributed. Suppose that Assumption 1 holds.*

$$m \geq \max\left\{\frac{16n^2d^2C_{\psi,d}}{C_0\lambda^2}\log\frac{4n^2}{\delta}, n^4\left(\frac{128\sqrt{2}d\sqrt{\mathcal{R}_{S,s_1}(\boldsymbol{\theta}(0))}}{(\lambda_{a,1} + \lambda_{\boldsymbol{\omega},1})\min\{\lambda_{a,2}, \lambda_{\boldsymbol{\omega},2}\}}2\log\frac{4m(d+1)}{\delta}\right)\right\}$$

*then with probability at least $1 - \delta$ over the choice of $\boldsymbol{\theta}(0)$, we have for any $t \in [t_1^*, t_2^*]$*

$$\mathcal{R}_{S,s_2}(\boldsymbol{\theta}(t)) \leq \mathcal{R}_{S,s_2}(\boldsymbol{\theta}(t_1^*))\exp\left(-\frac{t - t_1^*}{n}(\lambda_{a,2} + \lambda_{\boldsymbol{\omega},2})\right). \tag{16}$$

### 3.4 CONVERGENCE OF HRTA

By combining Propositions 2 and 5, we can establish the convergence of the HRTA.

**Theorem 2.** *Given $\delta \in (0, 1)$, $s_1 \in (0, +\infty)$, $\zeta > 1$ and the sample set $S = \{(\boldsymbol{x}_i, y_i)\}_{i=1}^n \subset \Omega$ with $\boldsymbol{x}_i$ 's drawn i.i.d. with uniformly distributed. Suppose that Assumption 1 holds,*

$$m \geq \max\left\{\frac{16n^2d^2C_{\psi,d}}{C_0\lambda^2}\log\frac{4n^2}{\delta}, n^4\left(\frac{128\sqrt{2}d\sqrt{\mathcal{R}_{S,s_1}(\boldsymbol{\theta}(0))}}{(\lambda_{a,1} + \lambda_{\boldsymbol{\omega},1})\min\{\lambda_{a,2}, \lambda_{\boldsymbol{\omega},2}\}}2\log\frac{4m(d+1)}{\delta}\right)\right\}$$

*then with probability at least $1 - \delta$ over the choice of $\boldsymbol{\theta}(0)$, we have*

$$\begin{cases} \mathcal{R}_{S,s_1}(\boldsymbol{\theta}(t)) \leq \mathcal{R}_{S,s_1}(\boldsymbol{\theta}(0)) \exp\left(-\frac{t}{n}(\lambda_{a,1} + \lambda_{\boldsymbol{\omega},1})\right), t \in [0, t_1^*] \\ \mathcal{R}_{S,s_2}(\boldsymbol{\theta}(t)) \leq \mathcal{R}_{S,s_2}(\boldsymbol{\theta}(t_1^*)) \exp\left(-\frac{t-t_1^*}{n}(\lambda_{a,2} + \lambda_{\boldsymbol{\omega},2})\right), t \in [t_1^*, t_2^*]. \end{cases}$$

*where $t_i^*$ are defined in Eqs. ($12$, $15$). Furthermore, we have that the decay speed in $[t_1^*, t_2^*]$ can be faster than $[0, t_1^*]$, i.e. $\exp\left(-\frac{t-t_1^*}{n}(\lambda_{a,2} + \lambda_{\boldsymbol{\omega},2})\right) \leq \exp\left(-\frac{t-t_1^*}{n}(\lambda_{a,1} + \lambda_{\boldsymbol{\omega},1})\right).$*

*Proof.* By amalgamating Propositions 2 and 5, we can readily derive the proof for Theorem 2. $\qquad\square$

**Corollary 1.** *Given $\delta \in (0,1)$, $s_1 \in (0, +\infty)$, $\zeta > 1$ and the sample set $S = \{(\boldsymbol{x}_i, y_i)\}_{i=1}^n \subset \Omega$ with $\boldsymbol{x}_i$'s drawn i.i.d. with uniformly distributed. Suppose that Assumption 1 holds,*

$$m \geq \max\left\{ \frac{16n^2 d^2 C_{\psi,d}}{C_0 \lambda^2} \log\frac{4n^2}{\delta}, n^4 \left( \frac{128\sqrt{2}d\sqrt{\mathcal{R}_{S,s_1}(\boldsymbol{\theta}(0))}}{(\lambda_{a,1} + \lambda_{\boldsymbol{\omega},1})\min\{\lambda_{a,2}, \lambda_{\boldsymbol{\omega},2}\}} 2\log\frac{4m(d+1)}{\delta} \right) \right\}$$

*and $s_2 := \inf\left\{ s \in (s_1, 2) \mid \mathcal{R}_{S,s}(\boldsymbol{\theta}(t_1^*)) > \zeta\mathcal{R}_{S,s_1}(\boldsymbol{\theta}(t_1^*)) \right\}$, then with probability at least $1 - \delta$ over the choice of $\boldsymbol{\theta}(0)$, we have for any $t \in [t_1^*, t_2^*]$*

$$\mathcal{R}_{S,s_2}(\boldsymbol{\theta}(t)) \leq \zeta\mathcal{R}_{S,s_1}(\boldsymbol{\theta}(0)) \exp\left(-\frac{t_1^*}{n}(\lambda_{a,1} + \lambda_{\boldsymbol{\omega},1})\right) \exp\left(-\frac{t-t_1^*}{n}(\lambda_{a,2} + \lambda_{\boldsymbol{\omega},2})\right). \quad (17)$$

**Remark 4.** *In the case where $M \geq 2$, note that we may consider the scenario where $m$ becomes larger. However, it's important to emphasize that the order of $m$ remains at $\mathcal{O}(n^4)$. This order does not increase substantially due to the fact that all the derivations presented in Subsection 3.3 can be smoothly generalized.*

Building upon the proof of Theorem 2, we can recognize the advantages of the HRTA. In the initial iteration, the training process does not differ significantly from training using training for pure ReLU networks. The traditional method can effectively reduce the loss function within the set $\mathcal{N}_1(\boldsymbol{\theta}(0))$, defined as:

$$\mathcal{N}_1(\boldsymbol{\theta}(0)) := \left\{ \boldsymbol{\theta} \mid \|\boldsymbol{G}_1(\boldsymbol{\theta}) - \boldsymbol{G}_1(\boldsymbol{\theta}(0))\|_F \leq \frac{1}{4}(\lambda_{a,1} + \lambda_{\boldsymbol{\omega},1}) \right\}.$$

In other words, the traditional method can effectively minimize the loss function within the time interval $[0, t_1^*]$, where $t_1^* = \inf\{t \mid \boldsymbol{\theta}(t) \notin \mathcal{N}_1(\boldsymbol{\theta}(0))\}$. Outside of this range, the training speed may slow down significantly and take a long time to converge. However, the HRTA transits the training dynamics to a new kernel by introducing a new activation function. This allows training to converge efficiently within a new range of $\boldsymbol{\theta}$, defined as:

$$\mathcal{N}_2(\boldsymbol{\theta}(t_1^*)) := \left\{ \boldsymbol{\theta} \mid \|\boldsymbol{G}_2(\boldsymbol{\theta}) - \boldsymbol{G}_2(\boldsymbol{\theta}(t_1^*))\|_F \leq \frac{1}{4}(\lambda_{a,2} + \lambda_{\boldsymbol{\omega},2}) \right\},$$

if $\boldsymbol{G}_2(\boldsymbol{\theta}(t_1^*))$ is strictly positive definite. Furthermore, we demonstrate that the minimum eigenvalue of $\boldsymbol{G}_2(\boldsymbol{\theta}(t_1^*))$ surpasses that of $\boldsymbol{G}_1(\boldsymbol{\theta}(0))$ under these conditions, as indicated by Theorem 1. This implies that, rather than decaying, the training speed may actually increase. This is one of the important reasons why we believe that relaxation surpasses traditional training methods in neural network training. Furthermore, in this paper, we provide evidence that $\mathbf{G}_2(\boldsymbol{\theta}(t_1^*))$ indeed becomes strictly positive definite when the width of neural networks is sufficiently large. Building upon Proposition 4, we can see that the increasing smallest eigenvalue of Gram matrices in each iteration contributes to a higher likelihood of $\mathbf{G}_2(\boldsymbol{\theta}(t_1^*))$ becoming strictly positive definite.

In summary, HRTA offers three key advantages in training:

• It dynamically builds the activation function, allowing loss functions to resume their decay when the training progress slows down, all without compromising the accuracy of the approximation.

• It accelerates the decay rate by increasing the smallest eigenvalue of Gram matrices with each homotopy iteration. Consequently, it enhances the probability of Gram matrices becoming positive definite in each iteration, further improving the training process.

# 4 EXPERIMENTAL RESULTS FOR THE HOMOTOPY RELAXATION TRAINING ALGORITHM

In this section, we will use several numerical examples to demonstrate our theoretical analysis results.

### 4.1 FUNCTION APPROXIMATION BY HRTA

In the first part, our objective is to employ NNs to approximate functions of the form $\sin\left(2\pi \sum_{i=1}^{d} x_i\right)$ for both $d = 1$ and $d = 3$. We will compare the performance of the HRTA method with the Adam optimizer. We used 100 uniform grid points for $d = 1$ and 125 uniform grid points for $d = 3$. Additional experiment details are provided in Appendix A.5. The following Figures 2 and 3 showcase the results achieved using a two-layer neural network with 1000 nodes to approximate $\sin\left(2\pi \sum_{i=1}^{d} x_i\right)$ for both $d = 1$ and $d = 3$. We observe oscillations in the figures, which result from plotting the loss against iterations using a logarithmic scale. To mitigate these fluctuations, we decrease the learning rate, allowing the oscillations to gradually diminish during the later stages of training. It's worth noting that these oscillations occur in both the Adam and HRTA optimization algorithms and do not significantly impact the overall efficiency of HRTA.

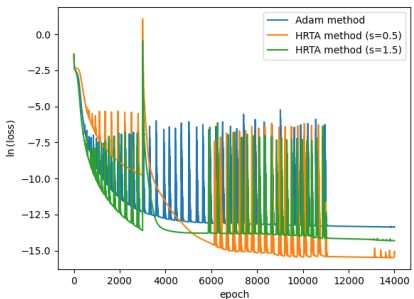

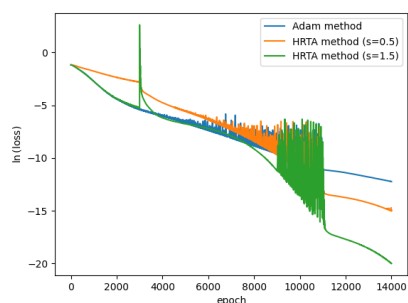

Figure 2: Approximation for $\sin(2\pi x)$

Figure 3: Approximation for $\sin(2\pi(x_1 + x_2 + x_3))$

In our approach, the HRTA method with $s = 0.5$ signifies that we initially employ $\sigma_{\frac{1}{2}}(x) := \frac{1}{2}\mathrm{Id}(x) + \frac{1}{2}\mathrm{ReLU}(x)$ as the activation functions. We transition to using ReLU as the activation function when the loss function does not decay rapidly. This transition characterizes the homotopy part of our method. Conversely, the HRTA kernel with $s = 1.5$ signifies that we begin with ReLU as the activation functions and switch to $\sigma_{\frac{3}{2}}(x) := -\frac{1}{2}\mathrm{Id} + \frac{3}{2}\mathrm{ReLU}$ as the activation functions when the loss function does not decay quickly. This transition represents the relaxation part of our method. Based on these experiments, it becomes evident that both cases, $s = 0.5$ and $s = 1.5$, outperform the traditional method in terms of achieving lower error rates. The primary driver behind this improvement is the provision of two opportunities for the loss function to decrease. While the rate of decay in each step may not be faster than that of the traditional method, as observed in the case of $s = 0.5$ for approximating $\sin(2\pi x)$ and $\sin(2\pi(x_1 + x_2 + x_3))$, it's worth noting that the smallest eigenvalue of the training dynamics is smaller than that of the traditional method when $s = 0.5$, as demonstrated in Theorem 1. This is the reason why, in the first step, it decays slower than the traditional method.

In the homotopy case with $s = 0.5$, it can be effectively utilized when we aim to train the ReLU neural network as the final configuration while obtaining a favorable initial value and achieving smaller errors than in the previous stages. Conversely, in the relaxation case with $s = 1.5$, it is valuable when we initially train the ReLU neural network, but the loss function does not decrease as expected. In this situation, changing the activation functions allows the error to start decreasing again without affecting the approximation quality. The advantage of both of these cases lies in their provision of two opportunities and an extended duration for the loss to decrease, which aligns with the results demonstrated in Theorem 2. This approach ensures robust training and improved convergence behavior in various scenarios.

Furthermore, our method demonstrates its versatility as it is not limited to very overparameterized cases or two-layer neural networks. We have shown its effectiveness even in the context of three-layer neural networks and other numbers of nodes (i.e., widths of 200, 400, and 1000). The error rates are summarized in Table 1, and our method consistently outperforms traditional methods

|      | Single Layer | | Multi Layer | |
|------|--------------|--------------|--------------|--------------|
|      | Adam | HRTA | Adam | HRTA |
| 200  | $2.02 \times 10^{-5}$ | $8.72 \times 10^{-7}$ (s=1.5) | $2.52 \times 10^{-7}$ | $9.47 \times 10^{-8}$ (s=0.5) |
| 400  | $6.54 \times 10^{-6}$ | $8.55 \times 10^{-7}$ (s=0.5) | $4.83 \times 10^{-8}$ | $1.8 \times 10^{-8}$ (s=1.5) |
| 1000 | $1.55 \times 10^{-6}$ | $1.88 \times 10^{-7}$ (s=0.5) | $2.20 \times 10^{-7}$ | $3.52 \times 10^{-9}$ (s=1.5) |

Table 1: Comparisons between HRTA and Adam methods on different NNs.

### 4.2 SOLVING PARTIAL DIFFERENTIAL EQUATION BY HRTA

In the second part, our goal is to solve the Poisson equation as follows:

$$\begin{cases} -\Delta u(x_1, x_2) = f(x_1, x_2) & \text{in } \Omega, \\ \frac{\partial u}{\partial \nu} = 0 & \text{on } \partial\Omega, \end{cases} \tag{18}$$

using HRTA. Here $\Omega$ is a domain within the interval $[0,1]^2$ and $f(x_1, x_2) = \pi^2 [\cos(\pi x_1) + \cos(\pi x_2)]$. The exact solution to this equation is denoted as $u^*(x_1, x_2) = \cos(\pi x_1) + \cos(\pi x_2)$. We still performed two iterations with 400 sample points and employed 1000 nodes. However, we used the activation function $\bar{\sigma}_{\frac{1}{2}}(x) = \frac{1}{2}\text{Id}(x) + \frac{1}{2}\bar{\sigma}(x)$, where $\bar{\sigma}(x) = \frac{1}{2}\text{ReLU}^2(x)$, which is smoother. Here we consider to solve partial differential equations by Deep Ritz method Yu & E (2018). In the deep Ritz method, the loss function of the Eq. (18) can be read as

$$\mathcal{E}_D(\boldsymbol{\theta}) := \frac{1}{2}\int_\Omega |\nabla\phi(\boldsymbol{x};\boldsymbol{\theta})|^2 \mathrm{d}\boldsymbol{x} + \frac{1}{2}\left(\int_\Omega \phi(\boldsymbol{x};\boldsymbol{\theta})\mathrm{d}\boldsymbol{x}\right)^2 - \int_\Omega f\phi(\boldsymbol{x};\boldsymbol{\theta})\mathrm{d}\boldsymbol{x},$$

where $\boldsymbol{\theta}$ represents all the parameters in the neural network. Here, $\Omega$ denotes the domain $(0,1)^d$. Proposition 1 in Lu et al. (2021) establishes the equivalence between the loss function $\mathcal{E}_D(\boldsymbol{\theta})$ and $\|\phi(\boldsymbol{x};\boldsymbol{\theta}) - u^*(\boldsymbol{x})\|_{H^1((0,1)^2)}$, where $u^*(\boldsymbol{x})$ denotes the exact solution of the PDEs which is $u^*(x_1, x_2) = \cos(\pi x_1) + \cos(\pi x_2)$, and $\|f\|_{H^1((0,1)^2)} := \left(\sum_{0 \leq |\alpha| \leq 1} \|D^{\boldsymbol{\alpha}}f\|_{L^2((0,1)^2)}^p\right)^{1/2}$. Therefore, we can use supervised learning via Sobolev training Czarnecki et al. (2017); Son et al. (2021); Vlassis & Sun (2021) to solve the Poisson equation efficiently and accurately. Our experiments reveal that HRTA remains effective even when $s = 1.5$, as demonstrated in Figures 4 and 5.

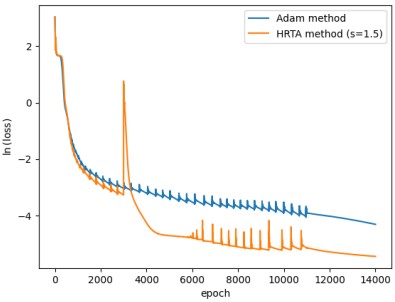
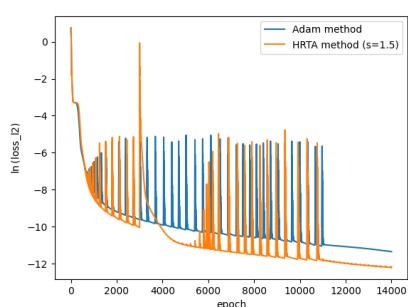

Figure 4: Loss function in Deep Ritz method

Figure 5: Solving Eq. (18) measured by $L^2$ norm

## 5 CONCLUSION

In summary, this paper introduces the Homotopy Relaxation Training Algorithm (HRTA), a method designed to expedite gradient descent when it encounters slowdowns. HRTA achieves this by relaxing homotopy activation functions to reshape the energy landscape of loss functions during slow convergence. Specifically, we adapt activation functions to boost the minimum eigenvalue of the gradient descent kernel, thereby accelerating convergence and increasing the likelihood of a positive minimum eigenvalue at each iteration. This paper establishes the theoretical basis for our algorithm, focusing on hyperparameters, while leaving the analysis in a more generalized context for future research.

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
