# A  APPENDIX

## A.1  PRELIMINARIES

### A.1.1  NEURAL NETWORKS

Let us summarize all basic notations used in the NNs as follows:

**1**. Matrices are denoted by bold uppercase letters. For example, $\boldsymbol{A} \in \mathbb{R}^{m \times n}$ is a real matrix of size $m \times n$ and $\boldsymbol{A}^\intercal$ denotes the transpose of $\boldsymbol{A}$. $\|\boldsymbol{A}\|_F$ is the Frobenius norm of the matrix $\boldsymbol{A}$.

**2**. Vectors are denoted by bold lowercase letters. For example, $\boldsymbol{v} \in \mathbb{R}^n$ is a column vector of size $n$. Furthermore, denote $\boldsymbol{v}(i)$ as the $i$-th elements of $\boldsymbol{v}$.

**3**. For a $d$-dimensional multi-index $\boldsymbol{\alpha} = [\alpha_1, \alpha_2, \cdots \alpha_d] \in \mathbb{N}^d$, we denote several related notations as follows:

$$
\begin{aligned}
&(a)\ |\boldsymbol{\alpha}| = |\alpha_1| + |\alpha_2| + \cdots + |\alpha_d|\,; \\
&(b)\ \boldsymbol{x}^\alpha = x_1^{\alpha_1} x_2^{\alpha_2} \cdots x_d^{\alpha_d},\ \boldsymbol{x} = [x_1, x_2, \cdots, x_d]^\intercal\,;
\end{aligned}
\tag{19}
$$

**4**. Assume $\boldsymbol{n} \in \mathbb{N}_+^n$, then $f(\boldsymbol{n}) = \mathcal{O}(g(\boldsymbol{n}))$ means that there exists positive $C$ independent of $\boldsymbol{n}, f, g$ such that $f(\boldsymbol{n}) \leq Cg(\boldsymbol{n})$ when all entries of $\boldsymbol{n}$ go to $+\infty$.

**5**. Define $\sigma(x) = \max\{0, x\}$ and $\sigma_s(x) = (1-s)\mathrm{Id}(x) + s\sigma(x)$ for $s > 0$. Two-layer NN structures are defined by:

$$
\phi_{s_p}(\boldsymbol{x}; \boldsymbol{\theta}) := \frac{1}{\sqrt{m}} \sum_{k=1}^m a_k \sigma_{s_p}(\boldsymbol{\omega}_k^\intercal \boldsymbol{x}).
\tag{20}
$$

**6**.

$$
\mathcal{R}_{S,s_p}(\boldsymbol{\theta}) := \frac{1}{2n} \sum_{i=1}^n |f(\boldsymbol{x}_i) - \phi_{s_p}(\boldsymbol{x}_i; \boldsymbol{\theta})|^2,
\tag{21}
$$

it is assumed that the sequence $\{\boldsymbol{x}_i\}_{i=1}^n$ consists of independent and identically distributed (i.i.d.) random variables. These random variables are uniformly distributed within the hypercube $(0,1)^d$, where $d$ is the dimension of the input space.

### A.1.2  RADEMACHER COMPLEXITY

In our further analysis, we will rely on the definition of Rademacher complexity and several lemmas related to it. Rademacher complexity is a fundamental concept in statistical learning theory and plays a crucial role in analyzing the performance of machine learning algorithms. It quantifies the complexity of a hypothesis class in terms of its ability to fit random noise in the data.

**Definition 1** (Rademacher complexity Anthony et al. (1999)). *Given a sample set $S = \{z_1, z_2, \ldots, z_M\}$ on a domain $\mathcal{Z}$, and a class $\mathcal{F}$ of real-valued functions defined on $\mathcal{Z}$, the empirical Rademacher complexity of $\mathcal{F}$ in $S$ is defined as*

$$
\mathrm{Rad}_S(\mathcal{F}) := \frac{1}{M} \mathbf{E}_{\Sigma_M} \left[ \sup_{f \in \mathcal{F}} \sum_{i=1}^M \tau_i f(z_i) \right],
$$

*where $\Sigma_M := \{\tau_1, \tau_2, \ldots, \tau_M\}$ are independent random variables drawn from the Rademacher distribution, i.e., $\mathbf{P}(\tau_i = +1) = \mathbf{P}(\tau_i = -1) = \frac{1}{2}$ for $i = 1, 2, \ldots, M$.*

**Lemma 3** (Rademacher complexity for linear predictors Shalev-Shwartz & Ben-David (2014)). *Let $\Theta = \{\boldsymbol{w}_1, \cdots, \boldsymbol{w}_m\} \in \mathbb{R}^d$. Let $\boldsymbol{G} = \{g(\boldsymbol{w}) = \boldsymbol{w}^\top \boldsymbol{x} : \|\boldsymbol{x}\|_1 \leq 1\}$ be the linear function class with parameter $\boldsymbol{x}$ whose $\ell^1$ norm is bounded by $1$. Then*

$$
\mathrm{Rad}_\Theta(\boldsymbol{G}) \leq \max_{1 \leq k \leq m} \|\boldsymbol{w}_k\|_\infty \sqrt{\frac{2\log(2d)}{m}}.
$$

**Lemma 4** (Rademacher complexity and generalization gap Shalev-Shwartz & Ben-David (2014)). *Suppose that $f$ in $\mathcal{F}$ are non-negative and uniformly bounded, i.e., for any $f \in \mathcal{F}$ and any $\boldsymbol{z} \in$*

$\mathcal{Z}, 0 \leq f(\boldsymbol{z}) \leq B$. *Then for any* $\delta \in (0, 1)$, *with probability at least* $1 - \delta$ *over the choice of* $n$ *i.i.d.* *random samples* $S = \{z_1, \ldots, z_n\} \subset \mathcal{Z}$, *we have*

$$\sup_{f \in \mathcal{F}} \left| \frac{1}{n} \sum_{i=1}^{n} f(\boldsymbol{z}_i) - \mathbf{E}_{\boldsymbol{z}} f(\boldsymbol{z}) \right| \leq 2 \operatorname{Rad}_S(\mathcal{F}) + 3B \sqrt{\frac{\log(4/\delta)}{2n}}.$$

## A.2 PROOF OF THEOREM 1

Before the proof, we need a lemma in the linear algebra.

**Lemma 5.** *Suppose* $\boldsymbol{A}$ *and* $\boldsymbol{B}$ *are strictly positive definite, we have that*

$$\lambda_{min}(\boldsymbol{A} + \boldsymbol{B}) \geq \lambda_{min}(\boldsymbol{A}) + \lambda_{min}(\boldsymbol{B}). \tag{22}$$

*Proof.* Let $\lambda_a$ be defined as $\lambda_{\min}(\boldsymbol{A})$ and $\lambda_b$ as $\lambda_{\min}(\boldsymbol{B})$. Consequently, we can assert that $\boldsymbol{A} + \boldsymbol{B} - (\lambda_a + \lambda_b)\boldsymbol{I}$ possesses positive definiteness. If we designate $\lambda$ as an eigenvalue of $\boldsymbol{A} + \boldsymbol{B}$, then it follows that $(\boldsymbol{A} + \boldsymbol{B})\boldsymbol{x}_* = \lambda \boldsymbol{x}_*$. This relationship can be expressed as:

$$(\boldsymbol{A} + \boldsymbol{B} - (\lambda_a + \lambda_b)\boldsymbol{I})\boldsymbol{x}_* = (\lambda - \lambda_a - \lambda_b)\boldsymbol{x}_*. \tag{23}$$

Consequently, we can deduce that $\lambda \geq \lambda_a + \lambda_b$, which further implies that $\lambda_{\min}(\boldsymbol{A} + \boldsymbol{B}) \geq \lambda_{\min}(\boldsymbol{A}) + \lambda_{\min}(\boldsymbol{B})$. □

*Proof of Theorem 1.* For the case $s_p < 1$, let's start by considering the expression for the matrix $\boldsymbol{K}_p^{[\boldsymbol{\omega}]}$ where

$$\boldsymbol{K}_p^{[\boldsymbol{\omega}]} = (K_{ij,p}^{[\boldsymbol{\omega}]})_{n \times n} = \left( \mathbf{E}_{(a,\boldsymbol{\omega})} a^2 \sigma'_{s_p}(\boldsymbol{\omega}^{\mathsf{T}} \boldsymbol{x}_i) \sigma'_{s_p}(\boldsymbol{\omega}^{\mathsf{T}} \boldsymbol{x}_j) \boldsymbol{x}_i \cdot \boldsymbol{x}_j \right)_{n \times n}. \tag{24}$$

Given the derivative of the activation function:

$$\sigma'_{s_p}(x) = \begin{cases} 1, & x > 0 \\ (1 - s_p), & x < 0 \\ 0, & x = 0 \end{cases} \quad \sigma'_{s_{p+1}}(x) = \begin{cases} 1, & x > 0 \\ (1 - s_{p+1}), & x < 0 \\ 0, & x = 0 \end{cases}$$

we have

$$\sigma'_{s_{p+1}}(x) = \sigma'_{s_p}(x) + (s_p - s_{p+1})\sigma(-x) \tag{25}$$

$$\mathbf{E}_{(a,\boldsymbol{\omega})} a^2 \sigma'_{s_{p+1}}(\boldsymbol{\omega}^{\mathsf{T}} \boldsymbol{x}_i) \sigma'_{s_{p+1}}(\boldsymbol{\omega}^{\mathsf{T}} \boldsymbol{x}_j) \boldsymbol{x}_i \cdot \boldsymbol{x}_j = \mathbf{E}_{(a,\boldsymbol{\omega})} a^2 \sigma'_{s_p}(\boldsymbol{\omega}^{\mathsf{T}} \boldsymbol{x}_i) \sigma'_{s_p}(\boldsymbol{\omega}^{\mathsf{T}} \boldsymbol{x}_j) \boldsymbol{x}_i \cdot \boldsymbol{x}_j$$

$$- (s_p - s_{p+1}) \mathbf{E}_{(a,\boldsymbol{\omega})} a^2 \left[ \sigma'(\boldsymbol{\omega}^{\mathsf{T}} \cdot (-\boldsymbol{x}_i)) \sigma'_{s_p}(\boldsymbol{\omega}^{\mathsf{T}} \boldsymbol{x}_j)(-\boldsymbol{x}_i) \cdot \boldsymbol{x}_j + \sigma'_{s_p}(\boldsymbol{\omega}^{\mathsf{T}} \boldsymbol{x}_i) \sigma'(\boldsymbol{\omega}^{\mathsf{T}} \cdot (-\boldsymbol{x}_j)) \boldsymbol{x}_i \cdot (-\boldsymbol{x}_j) \right]$$

$$+ (s_p - s_{p+1})^2 \mathbf{E}_{(a,\boldsymbol{\omega})} a^2 \sigma'(\boldsymbol{\omega}^{\mathsf{T}} \cdot (-\boldsymbol{x}_i)) \sigma'(\boldsymbol{\omega}^{\mathsf{T}} \cdot (-\boldsymbol{x}_j)) \boldsymbol{x}_i \cdot \boldsymbol{x}_j. \tag{26}$$

Furthermore, since

$$\sigma'(x) = \frac{\sigma'_{s_p}(x) - s_p \sigma'_{s_p}(-x)}{1 - s_p^2}, \tag{27}$$

we have

$$\sigma'(\boldsymbol{\omega}^{\mathsf{T}} \cdot (-\boldsymbol{x}_i)) \sigma'_{s_p}(\boldsymbol{\omega}^{\mathsf{T}} \boldsymbol{x}_j)(-\boldsymbol{x}_i) \cdot \boldsymbol{x}_j$$

$$= \frac{1}{1 - s_p^2} \left[ \sigma'_{s_p}(\boldsymbol{\omega}^{\mathsf{T}}(-\boldsymbol{x}_i)) \sigma'_{s_p}(\boldsymbol{\omega}^{\mathsf{T}} \boldsymbol{x}_j)(-\boldsymbol{x}_i) \boldsymbol{x}_j + s_p \sigma'_{s_p}(\boldsymbol{\omega}^{\mathsf{T}} \boldsymbol{x}_i) \sigma'_{s_p}(\boldsymbol{\omega}^{\mathsf{T}} \boldsymbol{x}_j) \boldsymbol{x}_i \boldsymbol{x}_j \right]$$

$$\sigma'_{s_p}(\boldsymbol{\omega}^{\mathsf{T}} \boldsymbol{x}_i) \sigma'(\boldsymbol{\omega}^{\mathsf{T}}(-\boldsymbol{x}_j)) \boldsymbol{x}_i \cdot (-\boldsymbol{x}_j)$$

$$= \frac{1}{1 - s_p^2} \left[ \sigma'_{s_p}(\boldsymbol{\omega}^{\mathsf{T}} \boldsymbol{x}_i) \sigma'_{s_p}(\boldsymbol{\omega}^{\mathsf{T}}(-\boldsymbol{x}_j))(-\boldsymbol{x}_i) \boldsymbol{x}_j + s_p \sigma'_{s_p}(\boldsymbol{\omega}^{\mathsf{T}} \boldsymbol{x}_i) \sigma'_{s_p}(\boldsymbol{\omega}^{\mathsf{T}} \boldsymbol{x}_j) \boldsymbol{x}_i \boldsymbol{x}_j \right]. \tag{28}$$

Therefore,

$$\boldsymbol{K}_{p+1}^{[\boldsymbol{\omega}]} = \left( 1 + \frac{2s_p(s_{p+1} - s_p)}{1 - s_p^2} \right) \boldsymbol{K}_p^{[\boldsymbol{\omega}]} + \frac{s_{p+1} - s_p}{1 - s_p^2}(\boldsymbol{M}_p^{[\boldsymbol{\omega}]} + \boldsymbol{H}_p^{[\boldsymbol{\omega}]}) + (s_{p+1} - s_p)^2 \boldsymbol{T}_M^{[\boldsymbol{\omega}]}. \tag{29}$$

When $s_p < 1$, with the initial condition $s_0 = 0$, we can establish the following inequalities based on Assumption 1, where $\boldsymbol{K}_0^{[\boldsymbol{\omega}]}, \boldsymbol{M}_0^{[\boldsymbol{\omega}]}, \boldsymbol{H}_0^{[\boldsymbol{\omega}]}$ is strictly positive, and Lemma 5 holds:

$$\lambda_{\min}(\boldsymbol{K}_1^{[\boldsymbol{\omega}]}) \geq 0. \tag{30}$$

The reason why $\boldsymbol{K}_0^{[\boldsymbol{\omega}]}, \boldsymbol{M}_0^{[\boldsymbol{\omega}]}, \boldsymbol{H}_0^{[\boldsymbol{\omega}]}$ are positive definite matrices is indeed attributed to the fact that $\sigma'_0(x)$ is a constant function. Specifically, for $\boldsymbol{K}_0^{[\boldsymbol{\omega}]}$, it can be represented as $(a(\boldsymbol{x}_1, \boldsymbol{x}_2, \ldots, \boldsymbol{x}_n))^{\mathsf{T}} a(\boldsymbol{x}_1, \boldsymbol{x}_2, \ldots, \boldsymbol{x}_n)$, which is inherently positive definite. Similar propositions can be derived for $\boldsymbol{M}_0^{[\boldsymbol{\omega}]}$ and $\boldsymbol{H}_0^{[\boldsymbol{\omega}]}$ based on the same principle.

Now, when $0 \leq s_p \leq s_{p+1}$ and $s_p < 1$ and Lemma 5 holds:

$$\lambda_{\min}(\boldsymbol{K}_{p+1}^{[\boldsymbol{\omega}]}) \geq \lambda_{\min}(\boldsymbol{K}_p^{[\boldsymbol{\omega}]}) \geq 0$$

due to Eqs. (29, 30).

For the case $s_p \geq 1$, we have that

$$\mathbf{E}_{(a,\boldsymbol{\omega})} a^2 \sigma'_{s_{p+1}}(\boldsymbol{\omega}^{\mathsf{T}} \boldsymbol{x}_i) \sigma'_{s_{p+1}}(\boldsymbol{\omega}^{\mathsf{T}} \boldsymbol{x}_j) \boldsymbol{x}_i \cdot \boldsymbol{x}_j = \mathbf{E}_{(a,\boldsymbol{\omega})} a^2 \sigma'_{s_p}(\boldsymbol{\omega}^{\mathsf{T}} \boldsymbol{x}_i) \sigma'_{s_p}(\boldsymbol{\omega}^{\mathsf{T}} \boldsymbol{x}_j) \boldsymbol{x}_i \cdot \boldsymbol{x}_j$$

$$-(s_p - s_{p+1}) \mathbf{E}_{(a,\boldsymbol{\omega})} a^2 \left[ \sigma'(\boldsymbol{\omega}^{\mathsf{T}} \cdot (-\boldsymbol{x}_i)) \sigma'_{s_p}(\boldsymbol{\omega}^{\mathsf{T}} \boldsymbol{x}_j)(-\boldsymbol{x}_i) \cdot \boldsymbol{x}_j + \sigma'_{s_p}(\boldsymbol{\omega}^{\mathsf{T}} \boldsymbol{x}_i) \sigma'(\boldsymbol{\omega}^{\mathsf{T}} \cdot (-\boldsymbol{x}_j)) \boldsymbol{x}_i \cdot (-\boldsymbol{x}_j) \right]$$

$$+(s_p - s_{p+1})^2 \mathbf{E}_{(a,\boldsymbol{\omega})} a^2 \sigma'(\boldsymbol{\omega}^{\mathsf{T}} \cdot (-\boldsymbol{x}_i)) \sigma'(\boldsymbol{\omega}^{\mathsf{T}} \cdot (-\boldsymbol{x}_j)) \boldsymbol{x}_i \cdot \boldsymbol{x}_j. \tag{31}$$

Furthermore, since

$$\sigma'_{s_p}(x) = \sigma'(x) + (1 - s_p)\sigma'(-x), \tag{32}$$

we have

$$\sigma'(\boldsymbol{\omega}^{\mathsf{T}} \cdot (-\boldsymbol{x}_i)) \sigma'_{s_p}(\boldsymbol{\omega}^{\mathsf{T}} \boldsymbol{x}_j)(-\boldsymbol{x}_i) \cdot \boldsymbol{x}_j$$

$$= \sigma'(\boldsymbol{\omega}^{\mathsf{T}} \cdot (-\boldsymbol{x}_i)) \sigma'(\boldsymbol{\omega}^{\mathsf{T}} \boldsymbol{x}_j)(-\boldsymbol{x}_i) \cdot \boldsymbol{x}_j - (1 - s_p)\sigma'(\boldsymbol{\omega}^{\mathsf{T}} \cdot (-\boldsymbol{x}_i)) \sigma'(\boldsymbol{\omega}^{\mathsf{T}}(-\boldsymbol{x}_j))(-\boldsymbol{x}_i) \cdot (-\boldsymbol{x}_j)$$

$$\sigma'_{s_p}(\boldsymbol{\omega}^{\mathsf{T}} \boldsymbol{x}_i) \sigma'(\boldsymbol{\omega}^{\mathsf{T}}(-\boldsymbol{x}_j)) \boldsymbol{x}_i \cdot (-\boldsymbol{x}_j)$$

$$= \sigma'(\boldsymbol{\omega}^{\mathsf{T}} \cdot \boldsymbol{x}_i) \sigma'(\boldsymbol{\omega}^{\mathsf{T}}(-\boldsymbol{x}_j))(-\boldsymbol{x}_i) \cdot \boldsymbol{x}_j - (1 - s_p)\sigma'(\boldsymbol{\omega}^{\mathsf{T}} \cdot (-\boldsymbol{x}_i)) \sigma'(\boldsymbol{\omega}^{\mathsf{T}}(-\boldsymbol{x}_j))(-\boldsymbol{x}_i) \cdot (-\boldsymbol{x}_j). \tag{33}$$

Therefore,

$$\boldsymbol{K}_{p+1}^{[\boldsymbol{\omega}]} = \boldsymbol{K}_p^{[\boldsymbol{\omega}]} - (1 - s_p)(s_{p+1} - s_p)(\boldsymbol{M}_M^{[\boldsymbol{\omega}]} + \boldsymbol{H}_M^{[\boldsymbol{\omega}]}) + (s_{p+1} - s_p)(s_{p+1} - s_p + 2)\boldsymbol{T}_M^{[\boldsymbol{\omega}]}.$$

When $0 \leq s_p \leq s_{p+1}$ and $s_p \geq 1$, we have that

$$\lambda_{\min}(\boldsymbol{K}_{p+1}^{[\boldsymbol{\omega}]}) \geq \lambda_{\min}(\boldsymbol{K}_p^{[\boldsymbol{\omega}]}) \geq 0$$

based on Assumption 1, as well as Lemma 5. Similar results can be derived for the Gram matrices with respect to the parameter $a$. □

### A.3 PROOFS IN $t_1$ ITERATION

*Proof of Lemma 1.* The proof can be found in (Luo et al., 2021, Lemma 9), for readable, we write the proof of this lemma here. Since $\mathbf{P}(|X| \leq B) \leq 2e^{-\frac{1}{2}B^2}$ if $X \sim \mathcal{N}(0, 1)$, we set $B = \sqrt{2 \log \frac{2m(d+1)}{\delta}}$ and obtain

$$\mathbf{P}\left(\max_{k \in [m]} \{|a_k(0)|, \|\boldsymbol{w}_k(0)\|_\infty\} > B\right) = \mathbf{P}\left(\max_{k \in [m], \alpha \in [d]} \{|a_k(0)|, |(w_k(0))_\alpha|\} > B\right)$$

$$= \mathbf{P}\left(\bigcup_{k=1}^{m} (|a_k(0)| > B) \bigcup \left(\bigcup_{\alpha=1}^{d} (|(w_k(0))_\alpha| > B)\right)\right)$$

$$\leq \sum_{k=1}^{m} \mathbf{P}(|a_k(0)| > B) + \sum_{k=1}^{m} \sum_{\alpha=1}^{d} \mathbf{P}(|(w_k(0))_\alpha| > B)$$

$$\leq 2me^{-\frac{1}{2}B^2} + 2mde^{-\frac{1}{2}B^2}$$

$$= 2m(d+1)e^{-\frac{1}{2}B^2}$$

$$= \delta.$$

□

*Proof of Lemma 2.* Let

$$\mathcal{G} := \{a\sigma_{s_1}(\boldsymbol{\omega}^\mathsf{T}\boldsymbol{x}), \boldsymbol{x} \in \Omega\} \tag{34}$$

and we have

$$|a(0)\sigma_{s_1}(\boldsymbol{\omega}^\mathsf{T}(0)\boldsymbol{x})| \leq 2d\log\frac{4m(d+1)}{\delta} =: B_1 \tag{35}$$

with probability at least $1 - \delta/2$ over the choice of $\boldsymbol{\theta}(0)$. Then we have

$$\sup_{\boldsymbol{x}\in\Omega}\left|\frac{1}{m}\sum_{k=1}^{m}a_k(0)\sigma_{s_1}\left(\boldsymbol{w}_k(0)\cdot\boldsymbol{x}\right)\right|$$

$$= \sup_{\boldsymbol{x}\in\Omega}\left|\frac{1}{m}\sum_{k=1}^{m}(a_k(0)\sigma_{s_1}\left(\boldsymbol{w}_k(0)\cdot\boldsymbol{x}\right) + B_1) - \left(\mathbf{E}_{(a,\boldsymbol{w})}a\sigma_{s_1}\left(\boldsymbol{w}^\top\boldsymbol{x}\right) + B_1\right)\right|$$

$$\leq 2\operatorname{Rad}_{\boldsymbol{\theta}(0)}(\boldsymbol{G}) + 12d\left(\log\frac{4m(d+1)}{\delta}\right)\sqrt{\frac{2\log(8/\delta)}{m}}$$

with probability at least $1 - \delta$ over the choice of $\boldsymbol{\theta}(0)$. The Rademacher complexity can be estimated by

$$\operatorname{Rad}_{\boldsymbol{\theta}(0)}(\boldsymbol{G}) = \frac{1}{m}\mathbf{E}_\tau\left[\sup_{\boldsymbol{x}\in\Omega}\sum_{k=1}^{m}\tau_k a_k(0)\sigma\left(\boldsymbol{w}_k(0)\cdot\boldsymbol{x}\right)\right]$$

$$\leq \frac{1}{m}\sqrt{2\log\frac{4m(d+1)}{\delta}}\mathbf{E}_\tau\left[\sup_{\boldsymbol{x}\in\Omega}\sum_{k=1}^{m}\tau_k\boldsymbol{w}_k(0)\cdot\boldsymbol{x}\right]$$

$$\leq \sqrt{2\log\frac{4m(d+1)}{\delta}}\sqrt{2d\log\frac{4m(d+1)}{\delta}}\frac{\sqrt{d}}{\sqrt{m}}$$

$$= \frac{2d\log\frac{4m(d+1)}{\delta}}{\sqrt{m}},$$

where the last inequality is a result of Lemma 1.

Therefore, we have

$$\sup_{\boldsymbol{x}\in\Omega}|\phi_{s_1}(\boldsymbol{x};\boldsymbol{\theta}(0))| \leq 2d\log\frac{4m(d+1)}{\delta}\left(2 + 6\sqrt{2\log(8/\delta)}\right) \tag{36}$$

and

$$\mathcal{R}_{S,s_1}(\boldsymbol{\theta}(0)) \leq \frac{1}{2}\left[1 + 2d\log\frac{4m(d+1)}{\delta}\left(2 + 6\sqrt{2\log(8/\delta)}\right)\right]^2. \tag{37}$$

$\square$

Next we are going to proof Proposition 1, before that, we need the definition of sub-exponential random variables and sub-exponential Bernstein's inequality.

**Definition 2** (Vershynin (2018)). *A random variable $X$ is sub-exponential if and only if its sub-exponential norm is finite i.e.*

$$\|X\|_{\psi_1} := \inf\{s > 0 \mid \mathbf{E}_X[e^{|X|/s} \leq 2.] \tag{38}$$

*Furthermore, the chi-square random variable $X$ is a sub-exponential random variable and $C_{\psi,d} := \|X\|_{\psi_1}$.*

**Lemma 6.** *Suppose that $\boldsymbol{w} \sim N(0, \boldsymbol{I}_d)$, $a \sim N(0,1)$ and given $\boldsymbol{x}_i, \boldsymbol{x}_j \in \Omega$. Then we have*

*(i) if $\mathrm{X} := \sigma_{s_1}\left(\boldsymbol{w}^\top\boldsymbol{x}_i\right)\sigma_{s_1}\left(\boldsymbol{x}\cdot\boldsymbol{x}_j\right)$, then $\|\mathrm{X}\|_{\psi_1} \leq dC_{\psi,d}$.*

*(ii) if $\mathrm{X} := a^2\sigma'_{s_1}\left(\boldsymbol{w}^\top\boldsymbol{x}_i\right)\sigma'_{s_1}\left(\boldsymbol{w}^\top\boldsymbol{x}_j\right)\boldsymbol{x}_i\cdot\boldsymbol{x}_j$, then $\|\mathrm{X}\|_{\psi_1} \leq dC_{\psi,d}$.*

*Proof.* The proof is similar with (Luo et al., 2021, Lemma 14).

(i) $|X| \leq d\|\boldsymbol{w}\|_2^2 = dZ$ and

$$
\begin{aligned}
\|X\|_{\psi_1} &= \inf\{s > 0 \mid \mathbf{E}_X \exp(|X|/s) \leq 2\} \\
&= \inf\{s > 0 \mid \mathbf{E}_{\boldsymbol{w}} \exp\left(\left|\sigma_{s_1}\left(\boldsymbol{w}^\top \boldsymbol{x}_i\right) \sigma_{s_1}\left(\boldsymbol{w}^\top \boldsymbol{x}_j\right)\right|/s\right) \leq 2\} \\
&\leq \inf\{s > 0 \mid \mathbf{E}_{\boldsymbol{w}} \exp\left(d\|\boldsymbol{w}\|_2^2/s\right) \leq 2\} \\
&= \inf\{s > 0 \mid \mathbf{E}_Z \exp(d|Z|/s) \leq 2\} \\
&= d\inf\{s > 0 \mid \mathbf{E}_Z \exp(|Z|/s) \leq 2\} \\
&= d\left\|\chi^2(d)\right\|_{\psi_1} \\
&\leq dC_{\psi,d}
\end{aligned}
$$

(ii) $|X| \leq d|a|^2 \leq dZ$ and $\|X\|_{\psi_1} \leq dC_{\psi,d}$. $\qquad\square$

**Theorem 3** (sub-exponential Bernstein's inequality Vershynin (2018)). *Suppose that* $X_1, \ldots, X_m$ *are i.i.d. sub-exponential random variables with* $\mathbf{E}X_1 = \mu$, *then for any* $s \geq 0$ *we have*

$$
\mathbf{P}\left(\left|\frac{1}{m}\sum_{k=1}^m X_k - \mu\right| \geq s\right) \leq 2\exp\left(-C_0 m \min\left(\frac{s^2}{\|X_1\|_{\psi_1}^2}, \frac{s}{\|X_1\|_{\psi_1}}\right)\right),
$$

*where* $C_0$ *is an absolute constant.*

*Proof of Proposition 1.* For any $\varepsilon > 0$, we define

$$
\begin{aligned}
\Omega_{ij,p}^{[a]} &:= \left\{\boldsymbol{\theta}(0) \mid \left|G_{ij,p}^{[a]}(\boldsymbol{\theta}(0)) - K_{ij,p}^{[a]}\right| \leq \frac{\varepsilon}{n}\right\} \\
\Omega_{ij,p}^{[\boldsymbol{\omega}]} &:= \left\{\boldsymbol{\theta}(0) \mid \left|G_{ij,p}^{[\boldsymbol{\omega}]}(\boldsymbol{\theta}(0)) - K_{ij,p}^{[\boldsymbol{\omega}]}\right| \leq \frac{\varepsilon}{n}\right\}.
\end{aligned}
\tag{39}
$$

Setting $\varepsilon \leq ndC_{\psi,d}$, by Theorem 3 and Lemma 6, we have

$$
\mathbf{P}(\Omega_{ij,p}^{[a]}) \geq 1 - 2\exp\left(-\frac{mC_0\varepsilon^2}{n^2 d^2 C_{\psi,d}}\right),
$$

$$
\mathbf{P}(\Omega_{ij,p}^{[\boldsymbol{\omega}]}) \geq 1 - 2\exp\left(-\frac{mC_0\varepsilon^2}{n^2 d^2 C_{\psi,d}}\right).
\tag{40}
$$

Therefore, with probability at least $\left[1 - 2\exp\left(-\frac{mC_0\varepsilon^2}{n^2 d^2 C_{\psi,d}^2}\right)\right]^{2n^2} \geq 1 - 4n^2\exp\left(-\frac{mC_0\varepsilon^2}{n^2 d^2 C_{\psi,d}^2}\right)$ over the choice of $\boldsymbol{\theta}(0)$, we have

$$
\left\|G_1^{[a]}(\boldsymbol{\theta}(0)) - K_1^{[a]}\right\|_F \leq \varepsilon
$$

$$
\left\|G_1^{[p]}(\boldsymbol{\theta}(0)) - K_1^{[p]}\right\|_F \leq \varepsilon.
\tag{41}
$$

Hence by taking $\varepsilon = \frac{\lambda_1}{4}$ and $\delta = 4n^2\exp\left(-\frac{mC_0\lambda_1^2}{16n^2 d^2 C_{\psi,d}^2}\right)$, where $\lambda_1 = \min\{\lambda_{a,1}, \lambda_{\boldsymbol{\omega},1}\}$

$$
\begin{aligned}
\lambda_{\min}\left(\boldsymbol{G}_1\left(\boldsymbol{\theta}(0)\right)\right) &\geq \lambda_{\min}\left(\boldsymbol{G}_1^{[a]}\left(\boldsymbol{\theta}(0)\right)\right) + \lambda_{\min}\left(\boldsymbol{G}_1^{[\boldsymbol{\omega}]}\left(\boldsymbol{\theta}(0)\right)\right) \\
&\geq \lambda_{a,1} + \lambda_{\boldsymbol{\omega},1} - \left\|G_1^{[a]}(\boldsymbol{\theta}(0)) - K_1^{[a]}\right\|_F - \left\|G_1^{[\boldsymbol{\omega}]}(\boldsymbol{\theta}(0)) - K_1^{[\boldsymbol{\omega}]}\right\|_F \\
&\geq \frac{3}{4}(\lambda_{a,1} + \lambda_{\boldsymbol{\omega},1}).
\end{aligned}
\tag{42}
$$

$\qquad\square$

*Proof of Proposition 2.* Due to Proposition 1 and the definition of $t_1^*$, we have that for any $\delta \in (0,1)$

$$
\lambda_{\min}\left(\boldsymbol{G}_1\left(\boldsymbol{\theta}\right)\right) \geq \frac{1}{2}(\lambda_{a,1} + \lambda_{\boldsymbol{\omega},1})
\tag{43}
$$

with probability at least $1 - \delta$ over the choice of $\boldsymbol{\theta}(0)$.

As we know

$$G_{ij,1} = G_{ij,1}^{[a]} + G_{ij,1}^{[\boldsymbol{\omega}]} = \sum_{k=1}^{m} \nabla_{a_k} \phi_{s_1}(\boldsymbol{x}_i; \boldsymbol{\theta}) \cdot \nabla_{a_k} \phi_{s_1}(\boldsymbol{x}_j; \boldsymbol{\theta}) + \frac{1}{m^2} \sum_{k=1}^{m} \nabla_{\boldsymbol{\omega}_k} \phi_{s_1}(\boldsymbol{x}_i; \boldsymbol{\theta}) \cdot \nabla_{\boldsymbol{\omega}_k} \phi_{s_1}(\boldsymbol{x}_j; \boldsymbol{\theta})$$
(44)

and

$$\begin{cases} \frac{\mathrm{d}a_k(t)}{\mathrm{d}t} = -\nabla_{a_k} \mathcal{R}_{S,s_1}(\boldsymbol{\theta}) = -\frac{1}{n\sqrt{m}} \sum_{i=1}^{n} e_{i,1} \sigma_{s_p} \left( \boldsymbol{w}_k^{\top} \boldsymbol{x}_i \right) \\ \frac{\mathrm{d}\boldsymbol{\omega}_k(t)}{\mathrm{d}t} = -\nabla_{\boldsymbol{w}_k} \mathcal{R}_{S,s_1}(\boldsymbol{\theta}) = -\frac{1}{n\sqrt{m}} \sum_{i=1}^{n} e_{i,1} a_i \sigma'_{s_p} \left( \boldsymbol{w}_k^{\top} \boldsymbol{x}_i \right) \boldsymbol{x}_i \end{cases}$$

where $e_{i,1} = |f(\boldsymbol{x}_i) - \phi_{s_1}(\boldsymbol{x}_i; \boldsymbol{\theta})|$.

Then finally we get that

$$\begin{aligned} \frac{\mathrm{d}}{\mathrm{d}t} \mathcal{R}_{S,s_1}(\boldsymbol{\theta}(t)) &= \sum_{k=1}^{m} \left( \nabla_{a_k} \mathcal{R}_{S,s_1}(\boldsymbol{\theta}) \frac{\mathrm{d}a_k(t)}{\mathrm{d}t} + \nabla_{\boldsymbol{\omega}_k} \mathcal{R}_{S,s_1}(\boldsymbol{\theta}) \frac{\mathrm{d}\boldsymbol{\omega}_k(t)}{\mathrm{d}t} \right) \\ &= -\sum_{k=1}^{m} \left( \nabla_{a_k} \mathcal{R}_{S,s_1}(\boldsymbol{\theta}) \nabla_{a_k} \mathcal{R}_{S,s_1}(\boldsymbol{\theta}) + \nabla_{\boldsymbol{\omega}_k} \mathcal{R}_{S,s_1}(\boldsymbol{\theta}) \nabla_{\boldsymbol{\omega}_k} \mathcal{R}_{S,s_1}(\boldsymbol{\theta}) \right) \\ &= -\frac{1}{n^2} \boldsymbol{e}_1^T \boldsymbol{G}_{ij,1}(\boldsymbol{\theta}(t)) \boldsymbol{e}_1 \\ &\leq -\frac{2}{n} \lambda_{\min} \left( \boldsymbol{G}_1 \left( \boldsymbol{\theta} \right) \right) \mathcal{R}_{S,s_1}(\boldsymbol{\theta}(t)) \\ &\leq -\frac{1}{n} (\lambda_{a,1} + \lambda_{\boldsymbol{\omega},1}) \mathcal{R}_{S,s_1}(\boldsymbol{\theta}(t)). \end{aligned}$$
(45)

Therefore,

$$\mathcal{R}_{S,s_1}(\boldsymbol{\theta}(t)) \leq \mathcal{R}_{S,s_1}(\boldsymbol{\theta}(0)) \exp \left( -\frac{t}{n} (\lambda_{a,1} + \lambda_{\boldsymbol{\omega},1}) \right).$$
(46)

$\square$

### A.4 PROOFS IN $t_2$ ITERATION

*Proof of Theorem 3.3.* For any $k \in [m]$, denote

$$\alpha(t) = \max_{k \in [m], s \in [0,t]} |a_k(s)|, \quad \omega(t) = \max_{k \in [m], s \in [0,t]} \|\boldsymbol{w}_k(s)\|_{\infty}$$

and we have

$$\left| \frac{\mathrm{d}a_k(t)}{\mathrm{d}t} \right|^2 = |\nabla_{a_k} \mathcal{R}_{S,s_1}(\boldsymbol{\theta})|^2 = \left| \frac{1}{n\sqrt{m}} \sum_{i=1}^{n} e_{i,1} \sigma_{s_p} \left( \boldsymbol{w}_k^{\top} \boldsymbol{x}_i \right) \right|^2 \leq \frac{2d^2 (\omega(t))^2 \mathcal{R}_{S,s_1}(\boldsymbol{\theta})}{m}.$$
(47)

Similarly, we have that

$$\left\| \frac{\mathrm{d}\boldsymbol{\omega}_k(t)}{\mathrm{d}t} \right\|_{\infty}^2 \leq \frac{2d^2 (\alpha(t))^2 \mathcal{R}_{S,s_1}(\boldsymbol{\theta})}{m}.$$

Due to the Proposition 2, we have

$$\begin{aligned} |a_k(t) - a_k(0)| &\leq \int_0^t |\nabla_{a_k} \mathcal{R}_{S,s_1}(\boldsymbol{\theta}(s))| \, \mathrm{d}s \\ &\leq \frac{\sqrt{2}d}{\sqrt{m}} \int_0^t \omega(s) \sqrt{\mathcal{R}_{S,s_1}(\boldsymbol{\theta}(s))} \mathrm{d}s \\ &\leq \frac{\sqrt{2}d}{\sqrt{m}} \omega(t) \int_0^t \sqrt{\mathcal{R}_{S,s_1}(\boldsymbol{\theta}(0))} \exp \left( -\frac{s}{2n} (\lambda_{a,1} + \lambda_{\boldsymbol{\omega},1}) \right) \mathrm{d}s \\ &\leq \frac{2\sqrt{2}nd\sqrt{\mathcal{R}_{S,s_1}(\boldsymbol{\theta}(0))}}{\sqrt{m}(\lambda_{a,1} + \lambda_{\boldsymbol{\omega},1})} \omega(t), \end{aligned}$$

with probability at least $1 - \delta/2$ over the choice of $\boldsymbol{\theta}(0)$. Similarly, we have

$$\|\boldsymbol{\omega}_k(t) - \boldsymbol{\omega}_k(0)\|_\infty \leq \frac{2\sqrt{2}nd\sqrt{\mathcal{R}_{S,s_1}(\boldsymbol{\theta}(0))}}{\sqrt{m}(\lambda_{a,1} + \lambda_{\boldsymbol{\omega},1})}\alpha(t). \tag{48}$$

Therefore, we have that

$$\alpha(t) \leq \alpha(0) + \frac{1}{\sqrt{m}}\kappa\omega(t)$$

$$\omega(t) \leq \omega(0) + \frac{1}{\sqrt{m}}\kappa\alpha(t)$$

where $\kappa = \frac{2\sqrt{2}nd\sqrt{\mathcal{R}_{S,s_1}(\boldsymbol{\theta}(0))}}{\lambda_{a,1} + \lambda_{\boldsymbol{\omega},1}}$. Therefore, when $m \geq \kappa^2$, we have

$$\max\{\alpha(t), \omega(t)\} \leq 2\alpha(0) + 2\omega(0).$$

Based on Lemma 1, with probability at least $1 - \delta/2$ over the choice of $\boldsymbol{\theta}(0)$ such that

$$\max_{k \in [m]}\{|a_k(0)|, \|\boldsymbol{\omega}_k(0)\|_\infty\} \leq \sqrt{2\log\frac{4m(d+1)}{\delta}}. \tag{49}$$

Therefore, we have

$$\max_{k \in [m]}\{|a_k(t) - a_k(0)|, \|\boldsymbol{\omega}_k(t) - \boldsymbol{\omega}_k(0)\|_\infty\} \leq \frac{8\sqrt{2}nd\sqrt{\mathcal{R}_{S,s_1}(\boldsymbol{\theta}(0))}}{\sqrt{m}(\lambda_{a,1} + \lambda_{\boldsymbol{\omega},1})}\sqrt{2\log\frac{4m(d+1)}{\delta}} \tag{50}$$

with probability at least $1 - \delta$ over the choice of $\boldsymbol{\theta}(0)$. $\square$

**Lemma 7.** *Suppose that $\boldsymbol{\omega} := \boldsymbol{\omega}(0) \sim N(0, \boldsymbol{I}_d), a = a(0) \sim N(0, 1)$ and given $\boldsymbol{x}_i, \boldsymbol{x}_j \in \Omega$. If*

$$m \geq \max\left\{\frac{16n^2d^2C_{\psi,d}}{C_0\lambda^2}\log\frac{4n^2}{\delta}, \frac{8n^2d^2\mathcal{R}_{S,s_1}(\boldsymbol{\theta}(0))}{(\lambda_{a,1} + \lambda_{\boldsymbol{\omega},1})^2}\right\}$$

*then with probability at least $1 - \delta$ over the choice of $\boldsymbol{\theta}(0)$, we have*

*(i) if $\mathrm{X} := \sigma_{s_2}(\bar{\boldsymbol{\omega}}^\top(\boldsymbol{\omega})\boldsymbol{x}_i)\sigma_{s_2}(\bar{\boldsymbol{\omega}}^\top(\boldsymbol{\omega}) \cdot \boldsymbol{x}_j)$, then $\|\mathrm{X}\|_{\psi_1} \leq 2dC_{\psi,d} + \frac{2d^2\psi(m)^2}{\log 2}$.*

*(ii) if $\mathrm{X} := \bar{a}(a)^2\sigma'_{s_2}(\bar{\boldsymbol{\omega}}^\top(\boldsymbol{\omega})\boldsymbol{x}_i)\sigma'_{s_2}(\bar{\boldsymbol{\omega}}^\top(\boldsymbol{\omega})\boldsymbol{x}_j)\boldsymbol{x}_i \cdot \boldsymbol{x}_j$, then $\|\mathrm{X}\|_{\psi_1} \leq 2dC_{\psi,d} + \frac{2d^2\psi(m)^2}{\log 2}$.*

*Proof.* (i)

$$|\mathrm{X}| \leq d\|\bar{\boldsymbol{\omega}}(\boldsymbol{\omega})\|_2^2 \leq 2d\|\boldsymbol{\omega}\|_2^2 + 2d\|\bar{\boldsymbol{\omega}}(\boldsymbol{\omega}) - \boldsymbol{\omega}\|_2^2 \leq 2d|Z| + 2d^2\psi(m)^2$$

and

$$\begin{aligned}
\|\mathrm{X}\|_{\psi_1} &= \inf\{s > 0 \mid \mathbf{E}_{\mathrm{X}}\exp(|\mathrm{X}|/s) \leq 2\} \\
&= \inf\{s > 0 \mid \mathbf{E}_{\boldsymbol{w}}\exp(|\sigma_{s_2}(\bar{\boldsymbol{\omega}}^\top(\boldsymbol{\omega})\boldsymbol{x}_i)\sigma_{s_2}(\bar{\boldsymbol{\omega}}^\top(\boldsymbol{\omega}) \cdot \boldsymbol{x}_j)|/s) \leq 2\} \\
&\leq \inf\left\{s > 0 \mid \mathbf{E}_{\boldsymbol{w}}\exp\left(\frac{2d|Z| + 2d^2\psi(m)^2}{s}\right) \leq 2\right\} \\
&\leq \inf\{s > 0 \mid \mathbf{E}_Z\exp(2d|Z|/s) \leq 2\} + \inf\left\{s > 0 \mid \mathbf{E}_{\boldsymbol{w}}\exp\left(\frac{2d^2\psi(m)^2}{s}\right) \leq 2\right\} \\
&= 2d\|\chi^2(d)\|_{\psi_1} + \frac{2d^2\psi(m)^2}{\log 2} \\
&\leq 2dC_{\psi,d} + \frac{2d^2\psi(m)^2}{\log 2}.
\end{aligned}$$

(ii) $|\mathrm{X}| \leq d|a|^2 \leq 2d|Z| + 2d^2\psi(m)^2$ and $\|\mathrm{X}\|_{\psi_1} \leq 2dC_{\psi,d} + \frac{2d^2\psi(m)^2}{\log 2}$. $\square$

To enhance simplicity and maintain consistent notation, we define:

$$C_{\psi,d,2} := 2C_{\psi,d} + \frac{2d\psi(m)^2}{\log 2}. \tag{51}$$

*Proof of Proposition 4.*

$$
\begin{aligned}
\bar{k}_2^{[a]}(\boldsymbol{x}, \boldsymbol{x}') :=& \mathbf{E}_{\boldsymbol{\omega}} \sigma_{s_2}\left(\bar{\boldsymbol{\omega}}^\top(\boldsymbol{\omega})\boldsymbol{x}\right)\sigma_{s_2}\left(\bar{\boldsymbol{\omega}}^\top(\boldsymbol{\omega})\cdot\boldsymbol{x}'\right) \\
\bar{k}_2^{[\boldsymbol{\omega}]}(\boldsymbol{x}, \boldsymbol{x}') :=& \mathbf{E}_{(a,\boldsymbol{\omega})}\bar{a}(a)^2\sigma'_{s_2}\left(\bar{\boldsymbol{\omega}}^\top(\boldsymbol{\omega})\boldsymbol{x}\right)\sigma'_{s_2}\left(\bar{\boldsymbol{\omega}}^\top(\boldsymbol{\omega})\boldsymbol{x}'\right)\boldsymbol{x}\cdot\boldsymbol{x}'.
\end{aligned}
\tag{52}
$$

The Gram matrices, denoted as $\bar{\boldsymbol{K}}_2^{[a]}$ and $\bar{\boldsymbol{K}}_2^{[\boldsymbol{\omega}]}$, corresponding to an infinite-width two-layer network with the activation function $\sigma_{s_2}$, can be expressed as follows:

$$
\begin{aligned}
\bar{K}_{ij,2}^{[a]} = \bar{k}_2^{[a]}(\boldsymbol{x}_i, \boldsymbol{x}_j),\ \bar{\boldsymbol{K}}_2^{[a]} = (\bar{K}_{ij,2}^{[a]})_{n\times n}, \\
\bar{K}_{ij,2}^{[\boldsymbol{\omega}]} = \bar{k}_2^{[\boldsymbol{\omega}]}(\boldsymbol{x}_i, \boldsymbol{x}_j),\ \bar{\boldsymbol{K}}_p^{[\boldsymbol{\omega}]} = (\bar{K}_{ij,2}^{[\boldsymbol{\omega}]})_{n\times n}.
\end{aligned}
\tag{53}
$$

The proof can be divided into two main parts. The first part, seeks to establish that the difference between $\boldsymbol{K}_2^{[a]} + \boldsymbol{K}_2^{[\boldsymbol{\omega}]}$ and $\bar{\boldsymbol{K}}_2^{[a]} + \bar{\boldsymbol{K}}_2^{[\boldsymbol{\omega}]}$ is small. In this case, the proof draws upon Proposition 3, which underscores the potential for the error in $\|\boldsymbol{\theta}(0) - \boldsymbol{\theta}(t^*)\|_\infty$ to be highly negligible when $m$ assumes a large value. The second part aims to demonstrate that the disparity between $\boldsymbol{G}(\boldsymbol{\theta}(t_1^*))$ and $\bar{\boldsymbol{K}}_2^{[a]} + \bar{\boldsymbol{K}}_2^{[\boldsymbol{\omega}]}$ is minimal. This particular proof relies on the application of sub-exponential Bernstein's inequality as outlined in Vershynin (2018) (Theorem 3).

First of all, we prove that the difference between $\boldsymbol{K}_2^{[a]} + \boldsymbol{K}_2^{[\boldsymbol{\omega}]}$ and $\bar{\boldsymbol{K}}_2^{[a]} + \bar{\boldsymbol{K}}_2^{[\boldsymbol{\omega}]}$ is small. Due to

$$
\begin{aligned}
\left|\bar{k}_2^{[a]}(\boldsymbol{x}, \boldsymbol{x}') - k_2^{[a]}(\boldsymbol{x}, \boldsymbol{x}')\right| \leq& \mathbf{E}_{\boldsymbol{\omega}}\left|\sigma_{s_2}\left(\bar{\boldsymbol{\omega}}^\top(\boldsymbol{\omega})\boldsymbol{x}\right)\sigma_{s_2}\left(\bar{\boldsymbol{\omega}}^\top(\boldsymbol{\omega})\boldsymbol{x}'\right) - \sigma_{s_2}\left(\boldsymbol{\omega}\boldsymbol{x}\right)\sigma_{s_2}\left(\boldsymbol{\omega}\cdot\boldsymbol{x}'\right)\right| \\
\leq& 2d\|\bar{\boldsymbol{\omega}}^\top(\boldsymbol{\omega}(0)) - \boldsymbol{\omega}(0)\|_\infty\|\boldsymbol{\omega}(0)\|_\infty \\
\leq& 2d\psi(m)\sqrt{2\log\frac{4m(d+1)}{\delta}}
\end{aligned}
\tag{54}
$$

with probability at least $1 - \delta$ over the choice of $\boldsymbol{\theta}(0)$. Therefore,

$$
\|\boldsymbol{K}_2^{[a]} - \bar{\boldsymbol{K}}_2^{[a]}\|_F \leq 2n\psi(m)\sqrt{2\log\frac{4m(d+1)}{\delta}}.
\tag{55}
$$

Similarly, we can obtain that

$$
\|\boldsymbol{K}_2^{[\boldsymbol{\omega}]} - \bar{\boldsymbol{K}}_2^{[\boldsymbol{\omega}]}\|_F \leq 2n\psi(m)\sqrt{2\log\frac{4m(d+1)}{\delta}}.
\tag{56}
$$

Set $\psi(m) \leq \frac{\min\{\lambda_{a,2}, \lambda_{\boldsymbol{\omega},2}\}}{16n\sqrt{2\log\frac{4m(d+1)}{\delta}}}$, i.e.

$$
m \geq n^4\left(\frac{128\sqrt{2}d\sqrt{\mathcal{R}_{S,s_1}(\boldsymbol{\theta}(0))}}{(\lambda_{a,1} + \lambda_{\boldsymbol{\omega},1})\min\{\lambda_{a,2}, \lambda_{\boldsymbol{\omega},2}\}}2\log\frac{4m(d+1)}{\delta}\right),
$$

we have

$$
\|\boldsymbol{K}_2^{[a]} - \bar{\boldsymbol{K}}_2^{[a]}\|_F, \|\boldsymbol{K}_2^{[\boldsymbol{\omega}]} - \bar{\boldsymbol{K}}_2^{[\boldsymbol{\omega}]}\|_F \leq \frac{1}{8}\min\{\lambda_{a,2}, \lambda_{\boldsymbol{\omega},2}\}.
$$

Furthermore, by sub-exponential Bernstein's inequality as outlined in Vershynin (2018) (Theorem 3), for any $\varepsilon > 0$, we define

$$
\begin{aligned}
\Omega_{ij,2}^{[a]} :=& \left\{\boldsymbol{\theta}(0) \mid \left|G_{ij,2}^{[a]}(\boldsymbol{\theta}(0)) - \bar{K}_{ij,2}^{[a]}\right| \leq \frac{\varepsilon}{n}\right\} \\
\Omega_{ij,2}^{[\boldsymbol{\omega}]} :=& \left\{\boldsymbol{\theta}(0) \mid \left|G_{ij,2}^{[\boldsymbol{\omega}]}(\boldsymbol{\theta}(0)) - \bar{K}_{ij,2}^{[\boldsymbol{\omega}]}\right| \leq \frac{\varepsilon}{n}\right\}.
\end{aligned}
\tag{57}
$$

Setting $\varepsilon \leq ndC_{\psi,d,2}$, by Theorem 3 and Lemma 6, we have

$$
\mathbf{P}(\Omega_{ij,2}^{[a]}) \geq 1 - 2\exp\left(-\frac{mC_0\varepsilon^2}{n^2d^2C_{\psi,d,2}}\right),
$$

$$
\mathbf{P}(\Omega_{ij,2}^{[\boldsymbol{\omega}]}) \geq 1 - 2\exp\left(-\frac{mC_0\varepsilon^2}{n^2d^2C_{\psi,d,2}}\right).
\tag{58}
$$

Therefore, with probability at least $\left[1 - 2\exp\left(-\frac{mC_0\varepsilon^2}{n^2d^2C_{\psi,d,2}^2}\right)\right]^{2n^2} \geq 1 - 4n^2\exp\left(-\frac{mC_0\varepsilon^2}{n^2d^2C_{\psi,d,2}^2}\right)$ over the choice of $\boldsymbol{\theta}(0)$, we have

$$\left\|G_2^{[a]}(\boldsymbol{\theta}(0)) - \bar{K}_2^{[a]}\right\|_F \leq \varepsilon$$
$$\left\|G_2^{[p]}(\boldsymbol{\theta}(0)) - \bar{K}_2^{[p]}\right\|_F \leq \varepsilon. \tag{59}$$

Hence by taking $\varepsilon = \frac{1}{8}\min\{\lambda_{a,2}, \lambda_{\boldsymbol{\omega},2}\}$ and $\delta = 4n^2\exp\left(-\frac{mC_0\lambda_1^2}{16n^2d^2C_{\psi,d,2}^2}\right)$, we obtain that

$$\begin{aligned}
\lambda_{\min}\left(\boldsymbol{G}_2\left(\boldsymbol{\theta}(t_1^*)\right)\right) \geq & \lambda_{\min}\left(\boldsymbol{G}_2^{[a]}\left(\boldsymbol{\theta}(t_1^*)\right)\right) + \lambda_{\min}\left(\boldsymbol{G}_2^{[\boldsymbol{\omega}]}\left(\boldsymbol{\theta}(t_1^*)\right)\right) \\
\geq & \lambda_{a,1} + \lambda_{\boldsymbol{\omega},1} - \left\|\boldsymbol{G}_2^{[a]}(\boldsymbol{\theta}(t_1^*)) - \bar{\boldsymbol{K}}_2^{[a]}\right\|_F - \left\|\boldsymbol{G}_2^{[\boldsymbol{\omega}]}(\boldsymbol{\theta}(t_1^*)) - \bar{\boldsymbol{K}}_2^{[\boldsymbol{\omega}]}\right\|_F \\
& - \|\boldsymbol{K}_2^{[a]} - \bar{\boldsymbol{K}}_2^{[a]}\|_F - \|\boldsymbol{K}_2^{[\boldsymbol{\omega}]} - \bar{\boldsymbol{K}}_2^{[\boldsymbol{\omega}]}\|_F \\
\geq & \frac{3}{4}(\lambda_{a,2} + \lambda_{\boldsymbol{\omega},2}). \tag{60}
\end{aligned}$$

$\square$

*Proof of Proposition 5.* Due to Proposition 4 and the definition of $t_2^*$, we have that for any $\delta \in (0,1)$

$$\lambda_{\min}\left(\boldsymbol{G}_2\left(\boldsymbol{\theta}(t)\right)\right) \geq \frac{1}{2}(\lambda_{a,1} + \lambda_{\boldsymbol{\omega},1}) \tag{61}$$

for any $t \in [t_1^*, t_2^*]$ with probability at least $1 - \delta$ over the choice of $\boldsymbol{\theta}(0)$.

As we know

$$G_{ij,2} = G_{ij,2}^{[a]} + G_{ij,2}^{[\boldsymbol{\omega}]} = \sum_{k=1}^m \nabla_{a_k}\phi_{s_2}(\boldsymbol{x}_i;\boldsymbol{\theta})\cdot\nabla_{a_k}\phi_{s_2}(\boldsymbol{x}_j;\boldsymbol{\theta}) + \frac{1}{m^2}\sum_{k=1}^m \nabla_{\boldsymbol{\omega}_k}\phi_{s_2}(\boldsymbol{x}_i;\boldsymbol{\theta})\cdot\nabla_{\boldsymbol{\omega}_k}\phi_{s_2}(\boldsymbol{x}_j;\boldsymbol{\theta}) \tag{62}$$

and

$$\begin{cases} \frac{\mathrm{d}a_k(t)}{\mathrm{d}t} = -\nabla_{a_k}\mathcal{R}_{S,s_2}(\boldsymbol{\theta}) = -\frac{1}{n\sqrt{m}}\sum_{i=1}^n e_{i,2}\sigma_{s_p}\left(\boldsymbol{w}_k^\top\boldsymbol{x}_i\right) \\ \frac{\mathrm{d}\boldsymbol{\omega}_k(t)}{\mathrm{d}t} = -\nabla_{\boldsymbol{w}_k}\mathcal{R}_{S,s_2}(\boldsymbol{\theta}) = -\frac{1}{n\sqrt{m}}\sum_{i=1}^n e_{i,2}a_i\sigma'_{s_p}\left(\boldsymbol{w}_k^\top\boldsymbol{x}_i\right)\boldsymbol{x}_i \end{cases}$$

where $e_{i,2} = |f(\boldsymbol{x}_i) - \phi_{s_2}(\boldsymbol{x}_i;\boldsymbol{\theta})|$.

Then finally we get that

$$\begin{aligned}
\frac{\mathrm{d}}{\mathrm{d}t}\mathcal{R}_{S,s_2}(\boldsymbol{\theta}(t)) &= \sum_{k=1}^m \left(\nabla_{a_k}\mathcal{R}_{S,s_2}(\boldsymbol{\theta})\frac{\mathrm{d}a_k(t)}{\mathrm{d}t} + \nabla_{\boldsymbol{\omega}_k}\mathcal{R}_{S,s_2}(\boldsymbol{\theta})\frac{\mathrm{d}\boldsymbol{\omega}_k(t)}{\mathrm{d}t}\right) \\
&= -\sum_{k=1}^m \left(\nabla_{a_k}\mathcal{R}_{S,s_2}(\boldsymbol{\theta})\nabla_{a_k}\mathcal{R}_{S,s_2}(\boldsymbol{\theta}) + \nabla_{\boldsymbol{\omega}_k}\mathcal{R}_{S,s_2}(\boldsymbol{\theta})\nabla_{\boldsymbol{\omega}_k}\mathcal{R}_{S,s_2}(\boldsymbol{\theta})\right) \\
&= -\frac{1}{n^2}\boldsymbol{e}_2^T\boldsymbol{G}_{ij,2}(\boldsymbol{\theta}(t))\boldsymbol{e}_2 \\
&\leq -\frac{2}{n}\lambda_{\min}\left(\boldsymbol{G}_2\left(\boldsymbol{\theta}\right)\right)\mathcal{R}_{S,s_2}(\boldsymbol{\theta}(t)) \\
&\leq -\frac{1}{n}(\lambda_{a,2} + \lambda_{\boldsymbol{\omega},2})\mathcal{R}_{S,s_2}(\boldsymbol{\theta}(t)). \tag{63}
\end{aligned}$$

Therefore,

$$\mathcal{R}_{S,s_2}(\boldsymbol{\theta}(t)) \leq \mathcal{R}_{S,s_2}(\boldsymbol{\theta}(t_1^*))\exp\left(-\frac{t - t_1^*}{n}(\lambda_{a,2} + \lambda_{\boldsymbol{\omega},2})\right). \tag{64}$$

$\square$

A.5 Experimental Details for the HRTA

A.5.1 Function approximation using supervised learning

**Example 1** (Approximating $\sin(2\pi x)$). *In the first example, our goal is to approximate the function $\sin(2\pi x)$ within the interval $[0,1]$ using two-layer neural networks (NNs) and the HRTA. We will provide a detailed explanation of the training process for the case of $s = 0.5$, which corresponds to the homotopy training case. The training process is divided into two steps:*

*1. In the first step, we employ the following approximation function:*

$$\phi_{\frac{1}{2}}(x;\boldsymbol{\theta}) := \frac{1}{\sqrt{1000}}\sum_{k=1}^{1000} a_k \sigma_{\frac{1}{2}}(\omega_k x) \tag{65}$$

*to approximate the function $\sin(2\pi x)$. Here, $\sigma_{\frac{1}{2}}(x) = \frac{1}{2}Id(x) + \frac{1}{2}\sigma(x)$, and the initial values of the parameters are drawn from a normal distribution $\boldsymbol{\theta} \sim \mathcal{N}(\mathbf{0}, \boldsymbol{I})$. We select random sample points (or grid points) $\{x_i\}_{i=1}^{100}$, which are uniformly distributed in the interval $[0,1]$. The loss function in this step is defined as*

$$\mathcal{R}_{S,\frac{1}{2}}(\boldsymbol{\theta}) := \frac{1}{200}\sum_{i=1}^{100} |f(x_i) - \phi_{\frac{1}{2}}(x_i;\theta)|^2. \tag{66}$$

*Therefore, we employ the Adam optimizer to train this model over 3000 steps to complete the first step of the process.*

*2. In the second step, we employ the following approximation function:*

$$\phi(x;\boldsymbol{\theta}) := \frac{1}{\sqrt{1000}}\sum_{k=1}^{1000} a_k \sigma(\omega_k x) \tag{67}$$

*to approximate the function $\sin(2\pi x)$. Here the initial values of the parameters are the results in the first step. The loss function in this step is defined as*

$$\mathcal{R}_S(\boldsymbol{\theta}) := \frac{1}{200}\sum_{i=1}^{100} |f(x_i) - \phi(x_i;\boldsymbol{\theta})|^2. \tag{68}$$

*Therefore, we employ the Adam optimizer to train this model over 13000 steps to complete the second step of the process and finish the training.*

*For the purpose of comparison, we employ a traditional method with the following approximation function:*

$$\phi(x;\boldsymbol{\theta}) := \frac{1}{\sqrt{1000}}\sum_{k=1}^{1000} a_k \sigma(\omega_k x) \tag{69}$$

*to approximate the function $\sin(2\pi x)$. Here, the initial values of the parameters are sampled from a normal distribution $\boldsymbol{\theta} \sim \mathcal{N}(\mathbf{0}, \boldsymbol{I})$. We select the same random sample points (or grid points) $x_{i_{i=1}^{100}}$ as used in the HRTA. The loss function in this step is defined as*

$$\mathcal{R}_S(\boldsymbol{\theta}) := \frac{1}{200}\sum_{i=1}^{100} |f(x_i) - \phi(x_i;\theta)|^2. \tag{70}$$

*Therefore, we employ the Adam optimizer to train this model over 16000 steps to complete the training.*

*In addition, we conducted experiments with neural networks that were not highly overparameterized, containing only 200 and 400 nodes. The results are illustrated in the following figures:*

**Example 2** (Approximating $\sin(2\pi(x_1 + x_2 + x_3))$). *The training methods in Example 1 and this current scenario share the same structure. The only difference is that in this case, all instances of $\omega$ and $x$ used in Example 1 have been extended to three dimensions. In Figure 3, we demonstrate that HRTA is effective in a highly overparameterized scenario, comprising 125 sample points with 1000 nodes. Additionally, we illustrate that HRTA remains effective in a scenario with less overparameterization, involving 400 nodes and 400 sample points. The results are presented below Figure 8.*

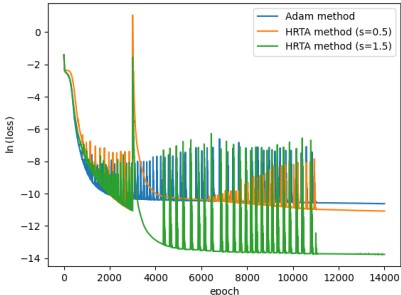 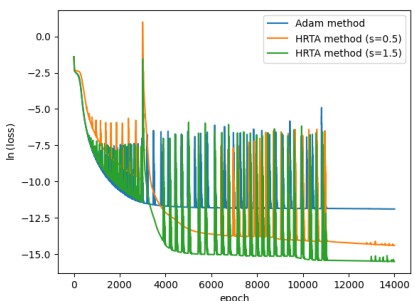

Figure 6: Approximation for $\sin(2\pi x)$ by NNs with 200 nodes

Figure 7: Approximation for $\sin(2\pi x)$ by NNs with 400 nodes

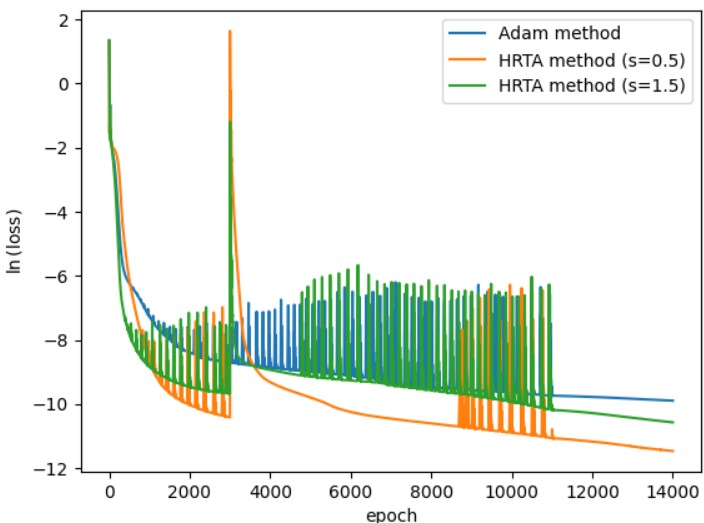

Figure 8: Approximation for $\sin(2\pi(x_1 + x_2 + x_3))$ with less overparameterization

### A.5.2 SOLVING PARTIAL DIFFERENTIAL EQUATIONS BY DEEP RITZ METHOD Yu & E (2018)

**Example 3.** *In this example, we aim to solve the Poisson equation given by:*

$$\begin{cases} -\Delta u(x_1, x_2) = \pi^2 \left[\cos(\pi x_1) + \cos(\pi x_2)\right] & in\ \Omega, \\ \frac{\partial u}{\partial \nu} = 0 & on\ \partial\Omega, \end{cases}$$

*by homotopy relaxation training methods, where $\Omega$ is a domain within the interval $[0, 1]^2$. The exact solution to this equation is denoted as $u^*(x_1, x_2) = \cos(\pi x_1) + \cos(\pi x_2)$.*

*1. In the first step, we employ the following approximation function:*

$$\bar{\phi}(\boldsymbol{x}; \boldsymbol{\theta}) := \frac{1}{\sqrt{1000}} \sum_{k=1}^{1000} a_k \bar{\sigma}(\boldsymbol{\omega}_k \boldsymbol{x}) \tag{71}$$

*to solve Passion equations. Here, $\bar{\sigma}(x) = \frac{1}{2}ReLU^2(x)$, and the initial values of the parameters are drawn from a normal distribution $\boldsymbol{\theta} \sim \mathcal{N}(\boldsymbol{0}, \boldsymbol{I})$. We select random sample points (or grid points) $\{x_i\}_{i=1}^{400}$, which are uniformly distributed in the interval $[0, 1]^2$. As per (Lu et al., 2021, Proposition 1), the loss function in the Deep Ritz method for solving this Poisson equation is indeed given by:*

$$\mathcal{R}_{S,\frac{1}{2}}(\boldsymbol{\theta}) := \frac{1}{800} \sum_{i=1}^{400} \left[ |u^*(\boldsymbol{x}_i) - \bar{\phi}(\boldsymbol{x}_i; \boldsymbol{\theta})|^2 + |\nabla u^*(\boldsymbol{x}_i) - \nabla\bar{\phi}(\boldsymbol{x}_i; \boldsymbol{\theta})|^2 \right]. \tag{72}$$

This loss function captures the discrepancy between the exact solution $u^*(\boldsymbol{x}_i)$ and the network's output $\bar{\phi}(\boldsymbol{x}_i; \boldsymbol{\theta})$, as well as the gradient of the exact solution and the gradient of the network's output, for each sampled point $\boldsymbol{x}_i$. Therefore, we employ the Adam optimizer to train this model over 16000 steps to complete the step.

*2. In the second step, we employ the following approximation function:*

$$\bar{\phi}_{\frac{3}{2}}(x; \boldsymbol{\theta}) := \frac{1}{\sqrt{1000}} \sum_{k=1}^{1000} a_k \bar{\sigma}(\omega_k x) \tag{73}$$

*to solve Possion equations. Here the initial values of the parameters are the results in the first step. The loss function in this step is defined as*

$$\mathcal{R}_S(\boldsymbol{\theta}) := \frac{1}{800} \sum_{i=1}^{400} \left[ |u^*(\boldsymbol{x}_i) - \bar{\phi}_{\frac{3}{2}}(\boldsymbol{x}_i; \boldsymbol{\theta})|^2 + |\nabla u^*(\boldsymbol{x}_i) - \nabla \bar{\phi}_{\frac{3}{2}}(\boldsymbol{x}_i; \boldsymbol{\theta})|^2 \right]. \tag{74}$$

*Therefore, we employ the Adam optimizer to train this model over 13000 steps to complete the step.*