# OpenReview forum: "Homotopy Relaxation Training Algorithms for Infinite-Width Two-Layer ReLU Neural Networks"
_ICLR.cc/2024/Conference — ICLR 2024 Conference Withdrawn Submission_

### Official Review · Reviewer_wUUw · 2023-11-01

**Soundness:** 2 fair
**Presentation:** 1 poor
**Contribution:** 1 poor
**Rating:** 3
**Confidence:** 3

**Summary:**

This work proposes algorithms to train infinite-width two-layer ReLU neural networks with convergence proof based on the neural tangent kernel with improved convergence rates in simple examples. This work claimed that its proposed method can be potentially applicable for other activation functions and neural networks (possibly deeper networks), but did not validate these claims.

**Strengths:**

Using a homotopy relaxation could be easily applicable for ReLU-based neural networks with mild modification.
Convergence proofs for this setting were presented.

**Weaknesses:**

Unfortunately, this manuscript missed so many prior works to investigate similar topics, which seriously undermine the novelty of the proposed method (see below).
This manuscript is very hard to read - no background review such as NTK and other related works, stating the algorithm first before presenting the settings, and so on.
Weak experimental comparisons (only with Adam).

**Questions:**

Q1. There are a number of works on training two-later ReLU NN or more that were not cited in this manuscript. See the following works and properly discuss, compare with them.
- R Arora et al., Understanding Deep Neural Networks with Rectified Linear Units, ICLR 2018
- Z Chen et al., A Generalized Neural Tangent Kernel Analysis for Two-layer Neural Networks, NeurIPS 2020
- G Yang & E Littwin, Tensor Programs IIb: Architectural Universality of Neural Tangent Kernel Training Dynamics, ICML 2021
- M Seleznova & G Kutyniok, Neural Tangent Kernel Beyond the Infinite-Width Limit: Effects of Depth and Initialization, ICML 2022
- T Gao et al., A global convergence theory for deep ReLU implicit networks via over-parameterization, ICLR 2022
These works also dealt with the training of neural networks using NTK for two-layer or more general networks. It is very critical to properly discuss prior works and clearly present / demonstrate the proposed method over them.

Q2. There are a number of claims without showing anything. For example, "It’s important to highlight that for s > 1 the decay speed may surpass that of a pure ReLU neural network." or "It is important to note that for cases with M > 2, all the analyses presented here can be readily extended." For me, they were not straightforward. Unclear claims should be clarified or be removed.

Q3. I am afraid that the applicability of this work is quite limited considering that modern deep learning heavily utilizes transformers and other advanced architectures.

Q4. Comparing only with Adam in the experiments seems limited. Note that Adam does not have to be the best for all cases. Comparing with other methods seems desirable.

---

### Official Review · Reviewer_4y1a · 2023-11-01

**Soundness:** 2 fair
**Presentation:** 3 good
**Contribution:** 2 fair
**Rating:** 3
**Confidence:** 4

**Summary:**

This paper presents a training approach called the Homotopy Relaxation Training Algorithm (HRTA) whose aim is to accelerate the training process in contrast to traditional methods, for two-layer (ReLU) networks only. Some theoretical and empirical results are shown.

**Strengths:**

The logic in the writing is clear.

**Weaknesses:**

1. Significance: Though the topic is very important, I am not sure how significant the results are, both theoretically and empirically. For instance, (Ergen and Pilanci, in AISTATS 2020) proved the convex geometry of two-layer ReLU networks, and based on this result, the acceleration of convergence using gradient based algorithms, like the proposed one, seems straightforward by modifying learning rates. I’d like to correct my comments if I am wrong.

2. Unconvincing empirical results: For instance,
(1) The tasks are not commonly used in evaluating the optimizers.
(2) The comparative optimizer is only Adam, even with no sgd that is most similar to the proposed algorithm.
(3) Learning rates are decreased in training, while in the algorithm there is no such procedure.
(4) No learning rate schedulers are compared.
Without such details, I hardly justify the correctness and fairness of the experiments.

**Questions:**

see my comments

---

### Official Review · Reviewer_kAtf · 2023-11-07

**Soundness:** 2 fair
**Presentation:** 2 fair
**Contribution:** 2 fair
**Rating:** 3
**Confidence:** 3

**Summary:**

In this work, the authors propose an activation function called the homotopy activation function, which is a linear combination of the identity function and a target activation function. Unlike a usual case using interpolation, extrapolation is allowed when gradually increasing the homotopy parameter. Using NTK theory, the authors show that the proposed method has faster convergence rates.

**Strengths:**

The overall construction of the so-called homotopy activation function appears to be interesting. The overall idea is more like gradually increasing the nonlinearity of the two-layer NN. Theoretical guarantees are provided for understand the convergence of the proposed activation function used with a two-layer NN.

**Weaknesses:**

The presentation must be improved significantly. Section 3 is very hard to follow and understand even with the assist of Figure 1. The notation is also not clearly defined. For instance, equations (20) and (21) should be given in the main text, and the definition of $\boldsymbol{\theta}$ is not clearly defined in (3).
The experiment settings also do not look very right to me. In the first experiment, the authors use a so-called two-step process with two values of $s$, i.e., 0.5/1.5 and 1. This is very different from what is claimed in the theory where the homotopy parameter $s$ is increased at each step. Also, the number of samples is just 100, which would be too small. It is also hard to disentangle the effect of the choice of $s$ and other hyperparameters. I would suggest the authors use vanilla SGD for comparison rather than Adam.

**Questions:**

It seems that when $s=0.5$, the homotopy activation function is equivalent to the so-called residual connection or skip connection. Could the theory explain how skip connections would help in training a two-layer NN?

Notation: Why writing $\mathrm{Id}(x)$ when it can be written as just $x$?
Typo: page 2: $\lbrace s_p\rbrace_{p=1}^M\in(0,2)$ is mathematically wrong. Both of them are sets.

page 8 before figures: $\sigma_{\frac32}(x) := -\frac12 \mathrm{Id}(x) + \frac32 \mathrm{ReLU}(x)$

---

### Author Response · Authors · 2023-11-16

Thank you to the reviewers and the ACs for their reviews. After consideration, we have decided to withdraw the manuscript.